# uPA-PAI-1 heteromerization promotes breast cancer progression by attracting tumorigenic neutrophils

Bernd Uhl[1,2], Laura A Mittmann[1,2], Julian Dominik[1,2], Roman Hennel[3], Bojan Smiljanov[1,2], Florian Haring[1,2], Johanna B Schaubächer[1,2], Constanze Braun[1,2], Lena Padovan[1,2], Robert Pick[2], Martin Canis[1], Christian Schulz[4], Matthias Mack[5], Ewgenija Gutjahr[6], Peter Sinn[6], Jörg Heil[7], Katja Steiger[8], Sandip M Kanse[9], Wilko Weichert[8,10], Markus Sperandio[11] [ID], Kirsten Lauber[3], Fritz Krombach[2] & Christoph A Reichel[1,2,*] [ID]

## Abstract

High intratumoral levels of urokinase-type plasminogen activator (uPA)-plasminogen activator inhibitor-1 (PAI-1) heteromers predict impaired survival and treatment response in early breast cancer. The pathogenetic role of this protein complex remains obscure. Here, we demonstrate that heteromerization of uPA and PAI-1 multiplies the potential of the single proteins to attract pro-tumorigenic neutrophils. To this end, tumor-released uPA-PAI-1 utilizes very low-density lipoprotein receptor and mitogen-activated protein kinases to initiate a pro-inflammatory program in perivascular macrophages. This enforces neutrophil trafficking to cancerous lesions and skews these immune cells toward a pro-tumorigenic phenotype, thus supporting tumor growth and metastasis. Blockade of uPA-PAI-1 heteromerization by a novel small-molecule inhibitor interfered with these events and effectively prevented tumor progression. Our findings identify a therapeutically targetable, hitherto unknown interplay between hemostasis and innate immunity that drives breast cancer progression. As a personalized immunotherapeutic strategy, blockade of uPA-PAI-1 heteromerization might be particularly beneficial for patients with highly aggressive uPA-PAI-1[high] tumors.

**Keywords** biomarker; breast cancer; fibrinolysis; innate immunity; neutrophils

**Subject Categories** Cancer; Immunology; Vascular Biology & Angiogenesis

## Introduction

Our immune system protects the organism from life-threatening infections and also from the development of malignant tumors. Importantly, however, distinct immune cell subsets are increasingly recognized to promote tumor initiation, progression, and metastasis formation in various cancer entities (Coffelt *et al*, 2016; Nicolas-Avila *et al*, 2017). In particular, the presence of neutrophilic granulocytes (neutrophils) in malignant lesions is associated with a very poor clinical outcome, despite representing a comparatively small leukocyte population in solid tumors (Gentles *et al*, 2015). In this regard, neutrophils are thought to support tumorigenesis *via* the production of cell injuring reactive oxygen species (ROS) and proteases, of pro-proliferative signals (*e.g.,* neutrophil elastase [NE]), of pro-angiogenic mediators (*e.g.,* matrix-metalloproteinase-9 [MMP-9], vascular endothelial growth factor [VEGF]), and of immunosuppressive factors (*e.g.,* arginase-1; Coffelt *et al*, 2016). In addition, neutrophils release net-like structures of DNA, histones, and other proteins ("neutrophil extracellular traps"; NETs) that facilitate tumor metastasis by sequestering circulating cancer cells (Demers *et al*, 2012; Cedervall *et al*, 2015). Interestingly, however,

1 Department of Otorhinolaryngology, University Hospital, Ludwig-Maximilians-Universität München, Munich, Germany
2 Walter Brendel Centre of Experimental Medicine, University Hospital, Ludwig-Maximilians-Universität München, Munich, Germany
3 Department of Radiation Oncology, University Hospital, Ludwig-Maximilians-Universität München, Munich, Germany
4 Department of Cardiology, University Hospital, Ludwig-Maximilians-Universität München, Munich, Germany
5 Department of Internal Medicine, University of Regensburg, Regensburg, Germany
6 Institute for Pathology, University of Heidelberg, Heidelberg, Germany
7 Department of Gynecology and Obstetrics, University of Heidelberg, Heidelberg, Germany
8 Department of Pathology, Technical University of Munich, Munich, Germany
9 Institute of Basic Medical Sciences, University of Oslo, Oslo, Norway
10 German Cancer Consortium (DKTK), partner site Munich, Ludwig-Maximilians-Universität München, Munich, Germany
11 Institute of Cardiovascular Physiology and Pathophysiology, Ludwig-Maximilians-Universität München, Munich, Germany
*Corresponding author. Tel: +49 89 4400 0; Fax: +49 89 2180 76538; E-mail: christoph.reichel@med.uni-muenchen.de

neutrophils have also been reported to exert anti-tumorigenic effects including antibody-dependent cell-mediated cytotoxicity (Coffelt et al, 2016). Accordingly, the existence of anti- ("N1") and pro-tumorigenic ("N2") phenotypes of neutrophils has been proposed (Fridlender et al, 2009). In contrast to these ambivalent functional properties of neutrophils in cancer, the mechanisms underlying the recruitment of these immune cells to malignant lesions remain largely unclear.

Fibrinolysis is a fundamental biological process that maintains tissue perfusion by preventing clot formation in blood vessels. Plasmin is the principal effector protease in the fibrinolytic system mediating the dissolution of fibrin polymers which is activated by proteolytic processing via the serine proteases uPA or tissue-plasminogen activator (tPA). The activity of these two plasminogen activators is tightly controlled by heteromerization with their inhibitor PAI-1. Besides their well-established role in fibrinolysis, it has been shown that the components of the fibrinolytic system are involved in additional biological processes including the regulation of cell adhesion, migration, and proliferation (Das et al, 2010; Smith & Marshall, 2010; Reichel et al, 2011b). In this context, plasmin, uPA, tPA, and PAI-1 have recently been demonstrated to promote neutrophil trafficking to sites of inflammation through their distinct proteolytic and non-proteolytic properties (Reichel et al, 2011a; Reichel et al, 2011b; Uhl et al, 2014; Praetner et al, 2018).

Breast cancer is the most frequently occurring oncological disorder in women worldwide (Carioli et al, 2017; Carioli et al, 2018). High intratumoral levels of uPA, PAI-1, and, especially, their heteromeric complexes have been identified as independent predictive factors for impaired survival (Andreasen et al, 1997; Schmitt et al, 1997; Knoop et al, 1998; Duffy et al, 1999; Foekens et al, 2000; Janicke et al, 2001; Sten-Linder et al, 2001; Look et al, 2002; Manders et al, 2004b) and treatment response (Harbeck et al, 2002a; Harbeck et al, 2002b; Manders et al, 2004a; Manders et al, 2004c) in early breast cancer, irrespective of the underlying histopathological subtype. Similarly, first clinical data suggest that high intratumoral expression of uPA-PAI-1 is related to poor survival rates in other malignancies such as pulmonary adenocarcinoma (Pappot et al, 2006). Although uPA-PAI-1 has been documented to exert a modest pro-proliferative and pro-migratory effect on cultured tumor cells (Webb et al, 1999; Webb et al, 2001), the precise function of this protein complex in the pathogenesis of (breast) cancer is still unknown.

# Results

With respect to the emerging role of neutrophils in cancer (Coffelt et al, 2016; Nicolas-Avila et al, 2017), the recently uncovered effects of uPA and PAI-1 on neutrophil trafficking (Reichel et al, 2011b; Praetner et al, 2018), and the clinical observations of uPA-PAI-1 heteromers as an independent prognosticator of impaired survival (Andreasen et al, 1997; Schmitt et al, 1997; Knoop et al, 1998; Duffy et al, 1999; Foekens et al, 2000; Janicke et al, 2001; Sten-Linder et al, 2001; Look et al, 2002; Manders et al, 2004b) and treatment response (Harbeck et al, 2002a; Harbeck et al, 2002b; Manders et al, 2004a; Manders et al, 2004c) in early breast cancer, we hypothesize that this protein complex supports the progression of this oncologic

disorder by promoting the trafficking of pro-tumorigenic neutrophils to malignant lesions.

## Effect of uPA-PAI-1 heteromerization on leukocyte trafficking

To prove this hypothesis, we first explored the effect of uPA-PAI-1 heteromerization on leukocyte trafficking. For this purpose, we examined the potential of the different components of the fibrinolytic system as well as of the heteromers of these proteins to attract circulating leukocytes by employing a mouse peritonitis assay and multi-channel flow cytometry. Intraperitoneal injection of recombinant mouse uPA, tPA, or PAI-1 induced a significant increase in numbers of neutrophils, classical monocytes, and non-classical monocytes, to the peritoneal cavity (Fig 1A). This increase was even more pronounced in animals receiving uPA-PAI-1 heteromers, whereas stimulation with tPA-PAI-1 heteromers did not initiate additional myeloid leukocyte responses as compared to the single proteins. Noteworthy, uPA-PAI-1 did not elicit the trafficking of B lymphocytes, CD4$^+$ T lymphocytes, or CD8$^+$ T lymphocytes to the peritoneal cavity. Moreover, uPA-PAI-1-dependent responses of classical monocytes were completely abolished in neutrophil-depleted animals (Fig 1B), collectively indicating that heteromerization of uPA and PAI-1 multiplies the potential of the single proteins to promote neutrophil trafficking.

To analyze the effect of uPA-PAI-1 heteromers on the trafficking of neutrophils (and classical monocytes) in more detail, we performed multi-channel in vivo microscopy in a cremaster muscle assay using CX$_3$CR-1$^{+/GFP}$ (monocyte reporter) mice. Although intrascrotal stimulation with uPA, PAI-1, or uPA-PAI-1 heteromers did not alter intravascular rolling of these innate immune cells, intravascular firm adherence and (subsequent) transmigration of Ly-6G$^+$ CX$_3$CR-1$^-$ neutrophils and Ly-6G$^-$ CX$_3$CR-1$^{low}$ classical monocytes (cMOs) to the perivascular tissue were significantly enhanced as compared to unstimulated controls (Fig 1C). In accordance with our previous findings, the potential of uPA-PAI-1 to induce these myeloid leukocyte responses was significantly higher as compared to the single proteins, but was similar as compared to the cytokine tumor necrosis factor (TNF; Appendix Fig S1A). Thus, uPA-PAI-1 heteromers potently mediate intravascular accumulation and subsequent extravasation of neutrophils to the perivascular space.

## Cell-specific effects of uPA-PAI-1 heteromers

Leukocyte trafficking from the microvasculature to their target destination is dependent on a complex multicellular interplay of these immune cells with perivascular macrophages and endothelial cells (Ley et al, 2007; Kolaczkowska & Kubes, 2013; Nourshargh & Alon, 2014). In a next step, we therefore sought to decipher cell-specific effects of uPA-PAI-1 heteromers. Employing immunostaining and confocal microscopy in cremasteric tissue whole mounts, uPA and PAI-1 were detected in the perivenular space of inflamed tissue as well as—to a lesser extent—on the microvascular endothelium of postcapillary venules (Fig 2A). To characterize the effect of extravascular uPA-PAI-1 heteromers on the activation of microvascular endothelial cells in vivo, these protein complexes were injected into the mouse scrotum. Here, uPA-PAI-1 induced a significant elevation in the expression of ICAM-1/CD54 and VCAM-1/CD106 on venular endothelial cells as compared to unstimulated controls

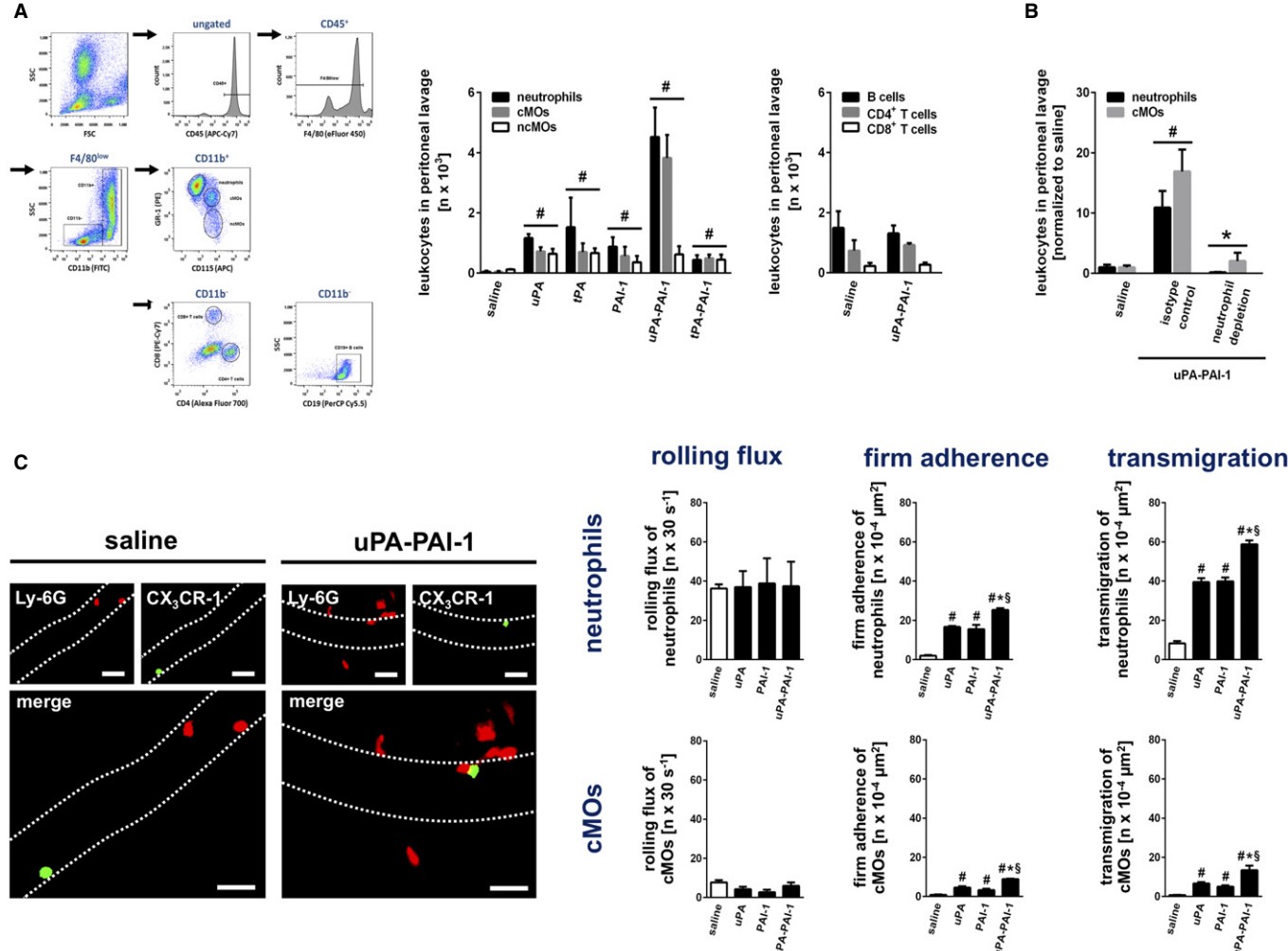

**Figure 1.  Effect of uPA-PAI-1 heteromerization on leukocyte trafficking.**

A   Trafficking of neutrophils (N), classical monocytes (cMOs), non-classical monocytes (ncMOs), B lymphocytes, CD4+ T lymphocytes, and CD8+ T lymphocytes to the
    peritoneal cavity as assessed 6 h after intraperitoneal stimulation with recombinant murine uPA, tPA, PAI-1, uPA-PAI-1, or tPA-PAI-1 in WT mice by multi-channel
    flow cytometry. The gating strategy and quantitative data are shown (mean ± SEM for n = 5 mice per group; #P < 0.05 vs. saline; one-way ANOVA).
B   Neutrophil and cMO trafficking to the peritoneal cavity elicited by recombinant murine uPA-PAI-1 as assessed in WT mice receiving neutrophil-depleting anti-Ly-6G
    mAbs or isotype control ABs, quantitative data are shown (mean ± SEM for n = 4 mice per group; #P < 0.05 vs. saline; *P < 0.05 vs. isotype; one-way ANOVA).
C   Intravascular endothelial cell interactions and transmigration of neutrophils (Ly-6G+ CX₃CR-1−; red) and cMOs (Ly-6G− CX₃CR-1low; green) to the perivascular tissue as
    assessed 6 h after intrascrotal stimulation with recombinant murine uPA, PAI-1, or uPA-PAI-1 in postcapillary venules of the cremaster of CX₃CR-1GFP/+ mice by multi-
    channel *in vivo* microscopy. Representative still images (scale bar: 50 μm) and quantitative data are shown (mean ± SEM for n = 4 mice per group; #P < 0.05 vs.
    saline; *P < 0.05 vs. uPA; §P < 0.05 vs. PAI-1; one-way ANOVA).

(Fig 2B). This effect of uPA-PAI-1 was more pronounced as compared to the single proteins (Appendix Fig S1B). In multiplex ELISA (Fig EV1A and B) and multi-channel flow cytometry (Fig 2C) *in vitro* analyses, we further identified that uPA-PAI-1 heteromers potently stimulate cultured mouse macrophages to produce a variety of CC and CXC chemokines as well as of cytokines including TNF. This uPA-PAI-1-elicited release of TNF was confirmed in primary mouse macrophages (Appendix Fig S2A). In contrast to TNF, however, exposure to uPA-PAI-1 heteromers did not directly activate cultured mouse microvascular endothelial cells *in vitro* as indicated by unchanged surface expression of ICAM-1/CD54 and VCAM-1/CD106 (Fig 2D). Consequently, co-culture with uPA-PAI-1

(or TNF)-stimulated RAW macrophages induced the surface expression of ICAM-1/CD54 and VCAM-1/CD106 in bEND.3 endothelial cells (Appendix Fig S2B). Hence, uPA-PAI-1 heteromers stimulate cytokine release from perivascular macrophages that, in turn, activates microvascular endothelial cells.

Intraluminal adherence to the microvascular endothelium of intravascularly rolling leukocytes is facilitated by interactions between endothelial members of the immunoglobulin superfamily (*e.g.*, ICAM-1/CD54, VCAM-1/CD106) and leukocyte β1 or β2 integrins in higher affinity conformation as elicited by endothelially presented chemokines (Ley *et al*, 2007; Kolaczkowska & Kubes, 2013; Nourshargh & Alon, 2014). Similar to chemokines, uPA and

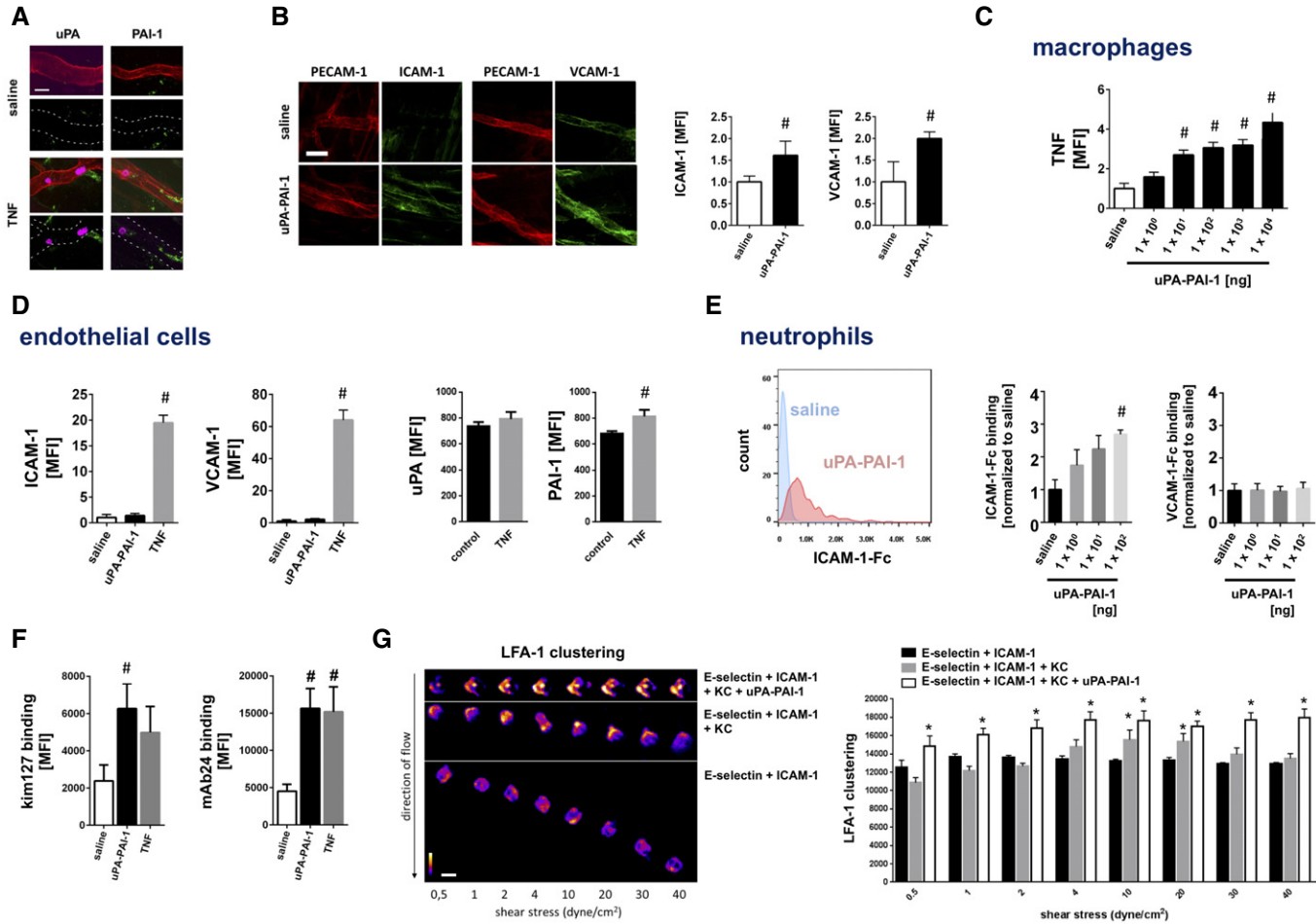

Figure 2. Mechanisms underlying uPA-PAI-1-dependent neutrophil trafficking.

A   Deposition of uPA or PAI-1 (green) in the cremaster muscle of WT mice as assessed separately *ex vivo* by confocal laser scanning microscopy 6 h after intrascrotal injection of TNF or saline, PECAM-1/CD31$^+$ postcapillary venules (red) and Ly-6G$^+$ neutrophils (purple) are depicted. Representative images are shown (scale bar: 50 μm).

B   Expression of PECAM-1/CD31 (red) and ICAM-1/CD54 or VCAM-1/CD106 (green) in the cremaster muscle of WT mice as assessed *ex vivo* by confocal laser scanning microscopy 6 h after intrascrotal injection of recombinant murine uPA-PAI-1 or saline, representative images (scale bar: 50 μm), and quantitative data are shown (mean ± SEM for $n = 4$ mice per group; $^\#P < 0.05$ vs. saline; $t$-test).

C, D   (C) Production of TNF was analyzed in mouse RAW 264.7 macrophages and (D) surface expression of ICAM-1/CD54, VCAM-1/CD106, uPA, and PAI-1 on mouse bEnd.3 microvascular endothelial cells as assessed *in vitro* by multi-channel flow cytometry upon exposure to recombinant murine uPA-PAI-1, TNF, or saline, quantitative data are shown (mean ± SEM for $n = 4$ experiments per group; $^\#P < 0.05$ vs. saline; one-way ANOVA/$t$-test).

E, F   (E) Binding of ICAM-1/CD54-Fc or VCAM-1/CD106-Fc to primary mouse neutrophils and (F) binding of conformation-specific mABs to primary human neutrophils ("mAB 24" recognizing high-affinity conformation, "kim127" recognizing intermediate and high-affinity conformation of β$_2$ integrins) as assessed upon exposure to recombinant murine/human uPA-PAI-1, TNF, or saline by multi-channel flow cytometry, a representative histogram plot and quantitative data are shown (mean ± SEM for $n = 4$–7 mice/human blood samples per group; $^\#P < 0.05$ vs. saline; one-way ANOVA).

G   Quantitative analysis of LFA-1/CD11a clustering on the surface of mouse neutrophils under increasing flow conditions *in vitro* in the presence of either E-selectin/CD62E + ICAM-1/CD54, E-selectin/CD62E + ICAM-1/CD54 + KC, or E-selectin/CD62E + ICAM-1/CD54 + KC + uPA-PAI-1 as assessed by spinning disk confocal microscopy, representative images and quantitative data are shown (scale bar: 10 μm; mean ± SD; $n = 3$ mice; $n = 10$–13 flow chambers, $n = 122$–222 cells); $^*P < 0.05$ vs. E-selectin/CD62E + ICAM-1/CD54; one-way ANOVA).

PAI-1 were also found on the endothelium of inflamed postcapillary venules (Fig 2A). Since activation of cultured microvascular endothelial cells did not substantially alter the surface expression of uPA or PAI-1 (Fig 2D), these circulating molecules are thought to be deposited on the surface of activated endothelial cells. In addition, uPA-PAI-1-elicited trafficking of neutrophils and classical monocytes to the peritoneal cavity were significantly reduced upon antibody blockade of the β1 integrin VLA-4/CD49d, the β2 integrins LFA-1/

CD11a or Mac-1/CD11b, or their counter receptors ICAM-1/CD54 or VCAM-1/CD106 (Fig EV2A). Consequently, endothelially deposited uPA-PAI-1 heteromers might also induce conformational changes in integrins on the surface of intravascularly rolling neutrophils, ultimately facilitating the adhesion of these innate immune cells to the inner vessel wall. In further experiments, uPA-PAI-1 heteromers (dose-dependently) increased the clustered binding of ICAM-1/CD54-Fc, but not of VCAM-1/CD106-Fc, to the surface of

neutrophils isolated from the peripheral blood of mice (Fig 2E; Appendix Fig S2C) which is indicative for the induction of higher affinity conformations in neutrophil β2 integrins. To directly evaluate the effect of uPA-PAI-1 heteromers on conformational changes in neutrophil β2 integrins, we also analyzed the binding of conformation-specific β2 integrin antibodies (only available in the human system) to human blood neutrophils. Exposure to uPA-PAI-1 led to enhanced recognition of the conformation-specific antibodies kim127 (indicating the presence of intermediate and high-affinity conformations of β2 integrins) and mAb24 (indicating the presence of the high-affinity conformation of β2 integrins; Fig 2F). Employing spinning disk confocal microscopy in an *in vitro* detachment assay under flow conditions, we further observed that uPA-PAI-1 also supports the clustering of β2 integrins such as lymphocyte function-associated integrin-1 (LFA-1/CD11a) on the surface of adhering murine neutrophils (Fig 2G). In summary, these data suggest that endothelially deposited uPA-PAI-1 heteromers promote neutrophil responses through effects on activation and surface clustering of β2 integrins.

## Mechanisms underlying uPA-PAI-1-dependent neutrophil responses

In fibrinolysis, the activity of uPA is controlled by heteromerization with its inhibitor PAI-1 which, in turn, allows for the endocytic clearance of these protein complexes from the circulation by endothelial receptors of the LDL receptor family such as very low-density lipoprotein receptor (VLDLr) or low-density lipoprotein receptor-related protein-1 (LRP-1) and subsequent activation of intracellular mitogen-activated protein kinases (MAPK) (Conese *et al*, 1995; Webb *et al*, 1999; Webb *et al*, 2001; Strickland *et al*, 2002). Consequently, uPA-PAI-1 heteromers might mediate neutrophil trafficking *via* such molecular events. Employing our peritonitis assay, antibody blockade of VLDLr, but not of LRP-1, as well as inhibition of downstream MAPK-dependent intracellular signaling pathways almost completely abolished uPA-PAI-1-elicited neutrophil recruitment to the peritoneal cavity (Fig EV2B). Moreover, these uPA-PAI-1-dependent neutrophil responses were not significantly altered when replacing native mouse uPA in the heteromers by human uPA (which does not bind to murine uPAR) or by DFP-uPA (in which the proteolytic activity of uPA is inhibited; Fig EV2C). In this context, multi-channel flow cytometry revealed that VLDLR is significantly stronger expressed on the surface of mouse neutrophils and macrophages as compared to LRP-1 (Fig EV3A). Accordingly, uPA-PAI-1-elicited TNF synthesis in macrophages (Fig EV3B) and ICAM-1/CD54-Fc binding (indicative for integrin affinity changes) in neutrophils (Fig EV3C) were significantly diminished upon antibody blockade of VLDLr or pharmacological inhibition of MAPK. Specifically, blocking antibodies directed against the N-terminal cysteine-rich domains 3–6 (clone 1H10), 1–2, and 5–6 (clone 1H5), but not against the cysteine-rich domains 7–8 (clone 5F3) of VLDLr (Yakovlev *et al*, 2016), as well as inhibitors of ERK and JNK MAPK, but not of p38 MAPK, significantly reduced these uPA-PAI-1-dependent processes in macrophages and neutrophils. These data suggest that uPA-PAI-1 heteromers mediate neutrophil trafficking through VLDLr and MAPK-dependent signaling pathways, but independently of the protease activity of uPA or the receptors LRP-1 and uPAR.

## Interrelation of uPA / PAI-1 expression, neutrophil infiltration, and disease outcome in human breast cancer

To explore the role of uPA-PAI-1 heteromers for neutrophil trafficking in human breast cancer, we analyzed a retrospective cohort of human breast cancer cases for neutrophilic infiltration (Appendix Table S1). In the tissue samples, uPA and PAI-1 protein expression (as assessed for clinical decision making during the time of treatment by the uPA/PAI-1 Femtelle assay (Schmitt *et al*, 2008)) positively correlated with neutrophil infiltration of low grade, but not of intermediate or high-grade tumors (Figs 3A and EV4). To complement these data, we studied RNA microarray data from the METABRIC breast cancer cohort (Curtis *et al*, 2012). Here, we found that in early, but not in advanced stages of disease, high RNA expression of PLAU and SERPINE1 (the genes encoding uPA and PAI-1) in the tumor is related to impaired overall survival of breast cancer patients (Figs 3B and EV5A and B). Importantly, this was not confounded by statistically significant enrichment of a distinct molecular subtype in the PLAU or SERPINE1 high expressing cases (Fig EV5C). Additionally, we observed a significant positive correlation between the RNA expression levels of PLAU and formyl peptide receptor 1 (FPR1), an established marker gene of neutrophils (Fig 3B). Thus, our results suggest that neutrophils attracted by uPA-PAI-1 to malignant lesions in human breast cancer are pro-tumorigenic.

## Phenotypic and functional properties of uPA-PAI-1-primed neutrophils

Neutrophils are supposed to adopt anti- and pro-tumorigenic properties according to their surface protein signatures (*e.g.*, NE, MMP-9, or VEGF) (Fridlender *et al*, 2009). Using multi-channel flow cytometry, we found that uPA-PAI-1-recruited neutrophils isolated from the peritoneal cavity of mice exhibit significantly higher levels of NE as compared to unstimulated neutrophils from the peripheral blood, whereas the surface expression of MMP-9 and VEGF remained unchanged (Fig 3C). To further characterize pro-tumorigenic properties of uPA-PAI-1-primed neutrophils, we employed different *in vitro* and *ex vivo* assays. Co-incubation of 4T1 breast cancer cells with uPA-PAI-1-primed neutrophils, but not direct exposure of uPA-PAI-1 protein to the tumor cells or blockade of uPA-PAI-1 heteromerization by a novel small-molecule inhibitor (WX-340), significantly increased the proliferation of 4T1 cells (Fig 3D; Appendix Fig S2D). This increase was significantly reduced upon application of a NE inhibitor. In contrast, neither co-incubation with uPA-PAI-1-primed neutrophils, exposure to uPA-PAI-1 protein, nor treatment with WX-340 altered the proliferation of cultured microvascular endothelial cells (Fig 3E). In addition, uPA-PAI-1 heteromers—unlike TNF—did not induce NET formation in neutrophils (Fig 3F), as evidenced by confocal microscopy on cremasteric tissue whole mounts. Hence, uPA-PAI-1-primed neutrophils exhibit distinct pro-tumorigenic properties that stimulate tumor cell proliferation *via* NE.

## Effect of inhibition of uPA-PAI-1 heteromerization on neutrophil trafficking and subsequent disease progression in 4T1 breast cancer

With respect to our present findings, we hypothesize that pharmacological inhibition of uPA-PAI-1 heteromerization interferes with

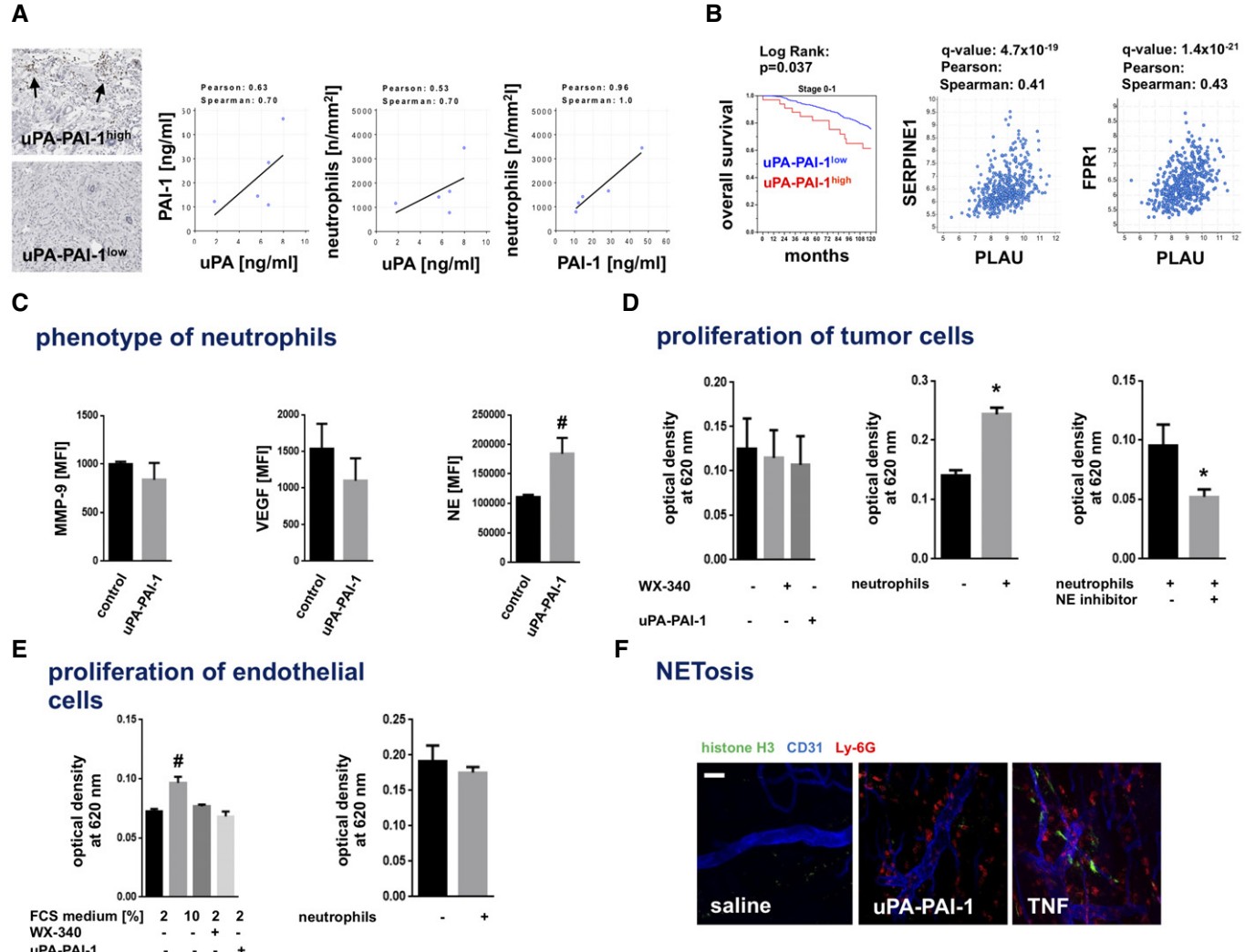

**Figure 3. Phenotypic and functional properties of uPA-PAI-1-primed neutrophils.**

A Correlation of uPA or PAI-1 protein expression and neutrophil infiltration in tumors as assessed by ELISA as well as immunohistochemistry and light microscopy in human breast cancer samples (histological grade: G1), representative images (scale bar: 100 μm) and quantitative data are shown.

B Correlation analyses of RNA expression levels of uPA (PLAU gene), PAI-1 (SERPINE1 gene), and the neutrophil marker gene FPR1, and overall survival of breast cancer patients (stages 0 and 1) with high and low PLAU or SERPINE1 gene expression levels (cutoff $z \geq 2.0$) in the METABRIC breast cancer cohort.

C Surface expression of MMP-9, VEGF, and NE as assessed on circulating neutrophils isolated from the peripheral blood of WT mice (saline) or from the peritoneal cavity of WT mice 6 h after intraperitoneal stimulation with uPA-PAI-1 (uPA-PAI-1) by multi-channel flow cytometry, quantitative data are shown (mean ± SEM for $n = 4$–6 mice per group; #$P < 0.05$ vs. saline; $t$-test).

D, E Proliferation of (D) 4T1 breast cancer cells or (E) bEnd.3 microvascular endothelial cells upon exposure to recombinant murine uPA-PAI-1, the uPA-PAI-1 inhibitor WX-340, or primary neutrophils isolated from the peritoneal cavity of WT mice undergoing 6 h of intraperitoneal stimulation with uPA-PAI-1 with or without addition of a NE inhibitor as assessed by a MTT assay, quantitative data are shown (mean ± SEM for $n = 3$ experiments per group; #$P < 0.05$ vs. neutrophils; *$P < 0.05$ vs. neutrophils + vehicle / #$P < 0.05$ vs. 2 % FCS medium; one-way ANOVA/$t$-test).

F Formation of NETs (histone H3[+]; green) as assessed ex vivo by confocal microscopy in the cremaster muscle of WT mice 6 h after intrascrotal injection of uPA-PAI-1, TNF, or saline, PECAM-1/CD31[+] postcapillary venules (blue) and Ly-6G[+] neutrophils (red) are depicted. Representative images are shown (scale bar: 50 μm).

pro-tumorigenic neutrophil responses in breast cancer and consequently attenuates tumor growth and metastasis. To prove this hypothesis, we first sought to characterize the effect of pharmacological inhibition of uPA and PAI-1 heteromerization on neutrophil trafficking. For this purpose, we used again the small-molecule inhibitor WX-340 that competitively and dose-dependently interferes with the binding of recombinant murine PAI-1 protein to uPA protein as evidenced in our ELISA analyses (Fig 4A). Using multi-channel *in vivo* microscopy on the mouse cremaster muscle, intrascrotal stimulation with TNF induced a significant elevation in numbers of intravascularly adherent and (subsequently) transmigrated neutrophils and classical monocytes (Fig 4B). This elevation was significantly diminished in animals treated with WX-340, indicating that blockade of uPA-PAI-1 heteromerization potently interferes with neutrophil trafficking to the inflamed perivascular space.

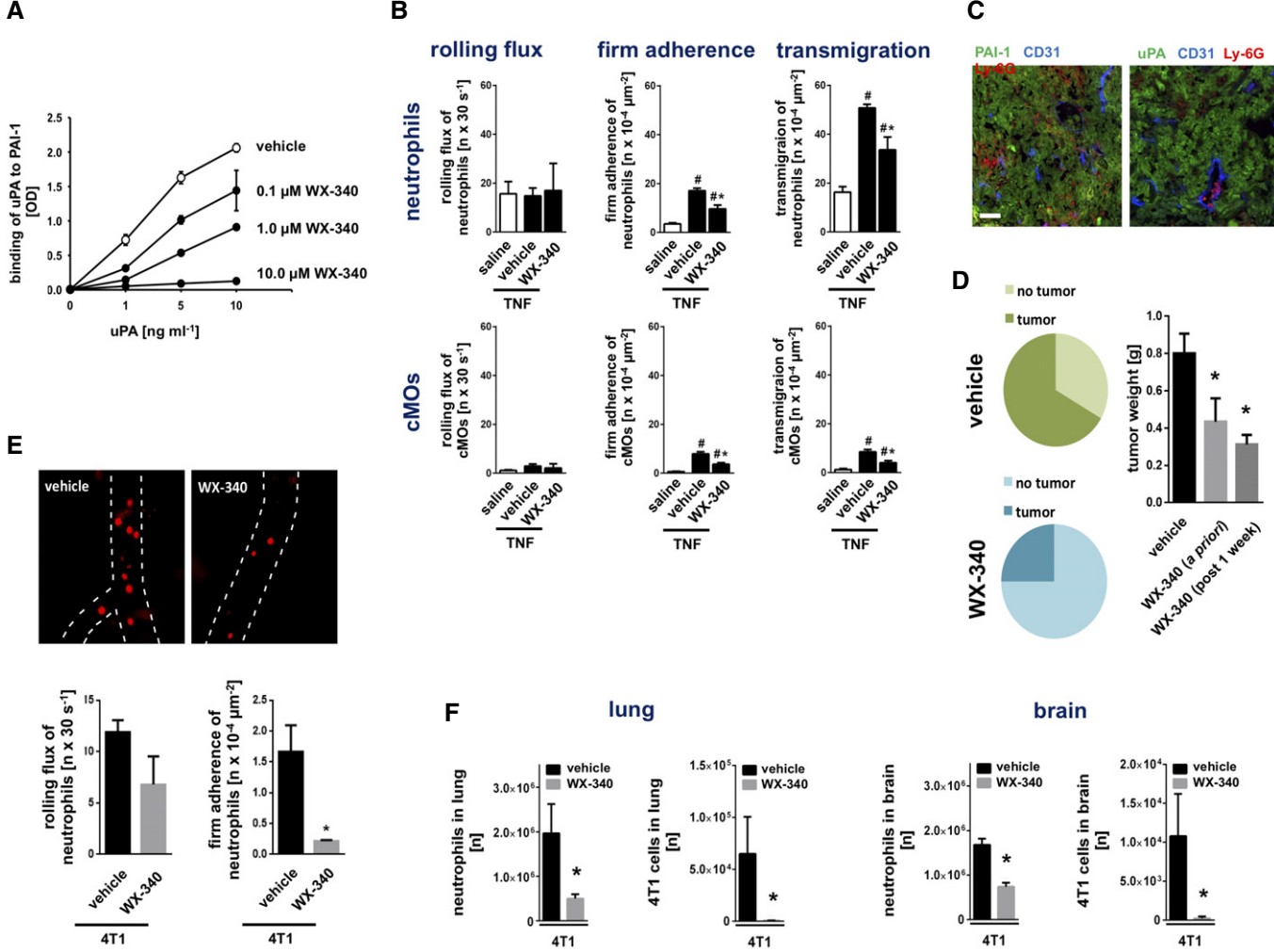

**Figure 4. Effect of WX-340 on experimental breast cancer progression and metastasis formation.**

A   Effect of compound WX-340 on binding of recombinant murine uPA to PAI-1 protein as assessed by ELISA, quantitative data are shown (mean ± SEM for $n = 3$ experiments per group; #$P < 0.05$ vs. drug vehicle; one-way ANOVA).

B   Intravascular rolling and firm adherence as well as transmigration of neutrophils and classical monocytes (cMOs) to the inflamed perivascular tissue as assessed 6 h after intrascrotal injection of TNF in postcapillary venules of the cremaster muscle of WT mice, quantitative data are shown (mean ± SEM for $n = 6$ mice per group; #$P < 0.05$ vs. saline; *$P < 0.05$ vs. drug vehicle; one-way ANOVA).

C   Expression of uPA or PAI-1 (green) in tumors as assessed in an orthotopic model of 4T1 breast cancer in WT mice by confocal microscopy, PECAM-1/CD31[+] postcapillary venules (blue) and Ly-6G[+] neutrophils (red) are depicted (scale bar: 100 μm).

D   Relative development rates and tumor weight in animals treated with WX-340 *a priori* or therapeutically after 1 week after tumor cell injection on a daily basis as assessed in an orthotopic model of 4T1 breast cancer in WT mice (mean ± SEM for $n = 4$–6 mice per group; *$P < 0.05$ vs. drug vehicle; one-way ANOVA).

E   Intravascular rolling and firm adherence of neutrophils as assessed in the tumor microvasculature 14 days after intradermal injection of 4T1 cells in the right ear of WT mice treated with WX-340 or vehicle by multi-channel *in vivo* microscopy, quantitative data are shown (mean ± SEM for $n = 4$–6 mice per group; *$P < 0.05$ vs. drug vehicle; *t*-test).

F   Numbers of neutrophils and 4T1 tumor cells in lungs and brain of WT mice receiving WX-340 or vehicle therapeutically after 1 week after tumor cell injection on a daily basis as assessed 14 days after intravenous injection of 4T1 tumor cells by multi-channel flow cytometry, quantitative data are shown (mean ± SEM for $n = 4$–6 mice per group; *$P < 0.05$ vs. drug vehicle; *t*-test).

To examine the effect of this compound on breast cancer progression, we established an orthotopic, syngeneic mouse model of 4T1 breast cancer which is well known for its strong intratumoral expression of uPA and PAI-1 (Fig 4C). In addition to tumor cells, infiltrated immune cells such as neutrophils also express these proteins (Reichel *et al*, 2011b; Praetner *et al*, 2018) (Appendix Fig S3A). Here, we demonstrate that both *a priori* and therapeutic treatment with WX-340 significantly diminished the local growth of these malignant tumors at early, pre-metastatic stages (Fig 4D; Appendix Fig S3B). In a heterotopic model of this oncological disorder in the auricle, we additionally document by multi-channel *in vivo* microscopy that WX-340 significantly reduced neutrophil trafficking to the neoplasms already on the level of intravascular adherence in the tumor microvaculature (Fig 4E). Conversely, depletion

of neutrophils almost completely abolished tumor growth, whereas depletion of cMOs only partially inhibited tumor progression in the orthotopic breast cancer model, further indicating that particularly neutrophils contribute to uPA-PAI-1-dependent tumorigenesis (Appendix Fig S3C and Appendix Table S2). To evaluate the effect of this inhibitor of uPA-PAI-1 heteromerization on metastasis formation in 4T1 breast cancer, we performed a final series of experiments. Similarly to our results for local tumor growth, both *a priori* and therapeutic treatment with WX-340 effectively attenuated metastasis formation and neutrophil recruitment to the malignant lesions in lungs and brain of the diseased mice (Fig 4E; Appendix Fig S4). Collectively, our data indicate that counteracting uPA-PAI-1 heteromerization by a small-molecule inhibitor effectively interferes with the trafficking of pro-tumorigenic neutrophils in experimental breast cancer and subsequently attenuates disease progression.

## Discussion

Clinical studies revealed that high intratumoral levels of uPA-PAI-1 heteromers predict impaired overall survival (Andreasen *et al*, 1997; Schmitt *et al*, 1997; Knoop *et al*, 1998; Duffy *et al*, 1999; Foekens *et al*, 2000; Janicke *et al*, 2001; Sten-Linder *et al*, 2001; Look *et al*, 2002; Manders *et al*, 2004b) and treatment response (Harbeck *et al*, 2002a; Harbeck *et al*, 2002b; Manders *et al*, 2004a; Manders *et al*, 2004c) in early breast cancer. The pathogenetic role of this protein complex, however, is still unclear. In line with previous reports (Reichel *et al*, 2011b; Uhl *et al*, 2014; Praetner *et al*, 2018), we here demonstrate that uPA, tPA, or PAI-1 induce the trafficking of neutrophils and—to a lesser extent—of classical monocytes. Most interestingly, heteromerization of uPA and PAI-1, but not of tPA and PAI-1, synergistically increased the potential of the single proteins to induce responses of these myeloid leukocytes. Notably, the migration of classical monocytes elicited by this protein complex was entirely dependent on the presence of neutrophils. In this context, we show that uPA-PAI-1 complexes mediate neutrophil extravasation very early on by triggering the intravascular adhesion of these innate immune cells, similar as the pro-inflammatory cytokine TNF. Importantly, uPA-PAI-1 heteromers did not induce detectable lymphocyte responses, collectively indicating that these protein complexes specifically promote the trafficking of neutrophils to their target destination.

Toward a more comprehensive understanding of these potent neutrophil-attracting properties of uPA-PAI-1 heteromers, we sought to identify the underlying mechanisms. Intravascular adherence and (subsequent) transmigration of neutrophils to the interstitial tissue requires interactions between endothelially expressed ICAM-1/CD54 or VCAM-1/CD106 and neutrophil β1 or β2 integrins in higher affinity conformations (Ley *et al*, 2007; Kolaczkowska & Kubes, 2013; Nourshargh & Alon, 2014). In our experiments, uPA and PAI-1 were found to be primarily present in the perivenular space of inflamed tissue, where the protein heteromers more potently induced surface expression of ICAM-1/CD54 and VCAM-1/CD106 on microvascular endothelial cells than the single proteins. However, we cannot clearly state to what extent intracellular and extracellular uPA-PAI-1 heteromerization occurs. In this context, uPA-PAI-1 did not directly activate microvascular endothelial cells, but initiated the release of various pro-inflammatory mediators in perivascular macrophages

including CC and CXC motif chemokines and the cytokine TNF. These macrophage-derived signals, in turn, induced surface expression of ICAM-1/CD54 and VCAM-1/CD106 on microvascular endothelial cells. In addition, circulating uPA and PAI-1 were deposited on the luminal surface of microvascular endothelial cells in inflamed tissue (Reichel *et al*, 2011b; Praetner *et al*, 2018). Subsequent complex formation of these proteins allowed them to induce higher affinity conformations of β2 integrins and supported the clustering of these adhesion molecules on the surface of adhering murine neutrophils. This process further strengthens the interactions of neutrophil β2 integrins and their endothelial counter receptor ICAM-1/CD54 and stabilizes the adhesion of these immune cells under increasing shear stress (Herter & Zarbock, 2013; Iwamoto & Calderwood, 2015; Ortega-Gomez *et al*, 2016), which is a prerequisite for the subsequent extravasation of neutrophils to the perivascular space.

To prevent hyperfibrinolysis, the fibrinolytic activity of uPA is antagonized by rapid, covalent binding to its inhibitor PAI-1 (Ellis *et al*, 1990). This exposes a cryptic binding site of PAI-1 that allows for high-affinity binding of the complexed protein (Croucher *et al*, 2007) to endothelially expressed receptors of the LDL receptor family including VLDLr and low-density lipoprotein receptor-related protein-1 (Conese *et al*, 1995; Webb *et al*, 1999; Strickland *et al*, 2002). As a consequence of such interactions, uPA-PAI-1 is endocytosed and induces sustained activation of intracellular MAPK (Webb *et al*, 2001), ultimately clearing this protein complex from the circulation. Nevertheless, the resulting effect of these molecular events on neutrophil trafficking is so far unknown. Here, we show that VLDLr, but not LRP-1, as well as MAPK-dependent intracellular signaling pathways mediate neutrophil recruitment to the peritoneal cavity elicited by uPA-PAI-1. The dispensability of LRP-1 in this context might be explained by the very low expression levels of this receptor protein on macrophages and neutrophils as compared to VLDLr. Accordingly, uPA-PAI-1-elicited cytokine synthesis in macrophages and integrin affinity changes in neutrophils were found to be facilitated through VLDLr as well as (subsequent) activation of ERK and JNK MAPK. Analyses with different site-specific blocking antibodies further revealed that the N-terminal cysteine-rich domains 3–6 (clone 1H10), 1–2, and 5–6 (clone 1H5), but not the cysteine-rich domains 7–8 (clone 5F3) of VLDLr (Yakovlev *et al*, 2016), are critical for mediating these uPA-PAI-1-dependent effects. Noteworthy, neither the protease activity of uPA nor its receptor uPAR were required for uPA-PAI-1-dependent neutrophil trafficking. This is in line with previous findings documenting that uPA mediates neutrophil responses *in vivo* independently of uPAR or its protease activity (Reichel *et al*, 2011b). Hence, uPA-PAI-1 promotes neutrophil trafficking under contribution of VLDLr and MAPK-dependent signaling events.

To elucidate the pathogenetic role of uPA-PAI-1-dependent neutrophil trafficking in breast cancer, we evaluated a retrospective cohort of breast cancer patients for neutrophilic infiltration. In this cohort, uPA and PAI-1 protein expression positively correlated with neutrophil infiltration of low grade, but not of intermediate or high-grade neoplasms. The missing link in tumors of higher histological grade might be due to the fact that neutrophilic infiltrations in these neoplasms are frequently triggered by necrosis which clearly follows different biological mechanisms (Munoz *et al*, 2017). In addition, analysis of RNA microarray data from the METABRIC breast cancer

cohort (Curtis *et al*, 2012) revealed that in early, but not in advanced stages of disease, RNA expression levels of the genes encoding uPA and PAI-1 are related to poor survival of breast cancer patients without enrichment of distinct molecular subtypes. Additionally, we observed a significant positive correlation between the RNA expression levels of the gene encoding uPA and a prominent marker gene of neutrophils, collectively suggesting that neutrophils attracted by uPA-PAI-1 exhibit pro-tumorigenic properties. In line with this assumption, we found that mouse neutrophils recruited by uPA-PAI-1 show higher expression levels of NE as compared to circulating neutrophils from the peripheral blood of unstimulated mice, which has been supposed (Fridlender *et al*, 2009) to point to a pro-tumorigenic phenotype of these immune cells. Furthermore, co-incubation of 4T1 breast cancer cells with uPA-PAI-1-primed neutrophils, but not direct exposure of uPA-PAI-1 protein to the tumor cells, potently increased the proliferation of 4T1 cells *in vitro*. This effect of uPA-PAI-1-primed neutrophils on the proliferation of breast cancer cells was largely dependent on NE which is thought to degrade insulin receptor substrate 1, a key regulator of phosphoinositide 3-kinase, and upregulate MAPK activity, ultimately stimulating tumor cell proliferation (Chen *et al*, 2004; Houghton *et al*, 2010). In contrast to our findings, however, uPA-PAI-1 heteromers have previously been reported to exhibit direct (*e.g.*, pro-proliferative) effects on MCF-7 and MDA-MB-435 breast cancer as well as HT 1080 fibrosarcoma cells by utilizing VLDLr (Webb *et al*, 1999; Webb *et al*, 2001). A possible explanation for these divergent results is the high degree of molecular heterogeneity of malignant tumor cells which might also include the expression of functional VLDLr. Importantly, uPA-PAI-1-primed neutrophils did not exhibit elevated surface expression levels of pro-angiogenic factors such as MMP-9 or VEGF as compared to circulating neutrophils and did not alter the proliferation of microvascular endothelial cells. Moreover, uPA-PAI-1 did not directly induce neutrophil extracellular trap (NET) formation in neutrophils sequestering circulating tumor cells (Demers *et al*, 2012; Cedervall *et al*, 2015), thus indicating that uPA-PAI-1 primes neutrophils toward a pro-tumorigenic phenotype that particularly exerts pro-proliferative effects on breast cancer cells *via* the release of NE.

To directly evaluate the functional relevance of uPA-PAI-1 heteromerization for the trafficking of pro-tumorigenic neutrophils in breast cancer, we tested the compound WX-340, a novel small-molecule inhibitor that competitively interferes with the binding of uPA to PAI-1. In accordance with our previous findings, this inhibitor effectively suppresses neutrophil adhesion to activated microvascular endothelium *in vivo*. To analyze the effect of this compound on the progression of malignant disease, we employed orthotopic and hetero-topic syngeneic models of 4T1 breast cancer in mice exhibiting a robust expression of uPA and PAI-1 in the tumors. *A priori* application of WX-340 severely compromised the progression of these malignant lesions *in vivo*, whereas this compound did not directly alter the proliferation of 4T1 tumor cells or microvascular endothelial cells *in vitro*. In these experiments, WX-340 affected the trafficking of circulating neutrophils to these highly aggressive neoplasms already on the level of intravascular adherence. Moreover, metastatic seeding of 4T1 tumors in lungs and brain as well as the associated neutrophil influx into the metastatic lesions were almost completely abrogated in inhibitor-treated animals. Most interestingly, even therapeutic application of WX-340 to animals with already established tumors and advanced stages of disease effectively interfered with local progression and metastatic tumor spread. Our findings extend previous observations of the anti-tumor and anti-metastatic activity of this compound in rat BN-472 mammary carcinoma and mouse HT1080 fibrosarcoma models (Setyono-Han *et al*, 2007).

In conclusion, our experimental data uncover a previously unrecognized biological role of uPA-PAI-1 heteromerization in breast cancer that potently promotes the trafficking of pro-tumorigenic neutrophils to malignant lesions. Counteracting this molecular process by a novel small-molecule inhibitor effectively interfered with pro-tumorigenic neutrophil responses and prevented advanced stages of this common oncological disorder in animal model systems. As an innovative personalized immunotherapeutic strategy, targeting this interplay between hemostasis and innate immunity might be particularly beneficial for patients with highly aggressive uPA-PAI-1[high] tumors in breast cancer.

# Materials and Methods

### Ethics

All animal experiments were approved by the local governmental authorities ("Regierung von Oberbayern") and conducted according to the guidelines to ensure animal welfare. During all surgical and experimental procedures, animals were anesthetized using ketamine (100 mg/kg; zoetis, Parsippany, New Jersey, USA) and xylazine (10 mg/kg; Bayer, Leverkusen, Germany).))

### Animals

Male C57BL/6 and female Balb/C mice were purchased from Charles River (Sulzfeld, Germany). Male $CX_3CR-1^{+/GFP}$ mice ("monocyte reporter mice") were generated as described previously (Jung *et al*, 2000) and backcrossed to the C57BL/6 background for 10 generations. $CX_3CR-1^{+/GFP}$ mice exhibit GFP[low] classical monocytes and GFP[high] non-classical monocytes.

Animals were housed under standard conditions ($22 \pm 2°C$, 30–60% humidity, 12 h light/dark cycle, lights on at 7 am) with access to food and water *ad libitum*. Experiments were performed with animals aging 6–8 weeks (body weight of 15–18 g).

### Cell lines

The murine mammary carcinoma cell line 4T1 was obtained from ATCC (Manassas, Virginia, USA) and stably transfected with ptd-Tomato-N1 (Clontech, Saint-Germain-en-Laye, France). Cells were cultured in RPMI-1640 media (Thermo Fisher Scientific, Waltham, Massachusetts, USA) media, supplemented with 10 % FBS (Biochrom, Berlin, Germany) and 1 % HEPES (PromoCell, Heidelberg, Germany) at 37°C and 5% $CO_2$. The murine endothelial cell line bEnd.3 and macrophage cell line RAW 264.7 were purchased from ATCC and cultured in DMEM (ATCC) supplemented with 10% FBS. Cells were routinely tested for mycoplasma contamination.

### Proteins

Recombinant murine high-molecular-weight uPA and PAI-1 as well as human uPA and PAI-1 (varying doses; Molecular Innovations,

Novi, MI) were used to induce leukocyte responses in different *in vitro* and *in vivo* assays (see below). In selected experiments, recombinant murine tumor necrosis factor (TNF; Abcam, Cambridge, UK) was used as positive control. For the generation of uPA-PAI-1 heteromers or DFP-uPA, murine uPA was titrated with murine PAI-1 or diisopropylfluorophosphate (DFP; Calbiochem, Darmstadt, Germany) so that no proteolytic activity remained.

## Inhibitors

The MAPK inhibitors FR180204 (ERK1/2 MAPK; 30 mg/kg body weight i.v.), SB202580 (p38 MAPK; 30 mg/kg body weight i.v.), or SP600125 (JNK MAPK; 30 mg/kg body weight i.v.; Sigma Aldrich GmbH, Taufkirchen, Germany) were used to characterize the functional relevance of MAPK for uPA-PAI-1-elicited neutrophil trafficking. A NE inhibitor (sivelestat; 150 μM; Sigma Aldrich) was used to evaluate the role of NE for 4T1 tumor cell proliferation. The competitive small-molecule WX-340 (10 mg/kg body weight i.p. for *in vivo* experiments; Heidelberg Pharma AG, Ladenburg, Germany) was used to inhibit heteromerization of uPA and PAI-1.

## Experimental groups in animal experiments

In first experiments, leukocyte recruitment to the peritoneal cavity of C57BL/6 mice was analyzed by multi-channel flow cytometry 6 h after intraperitoneal injection of recombinant murine uPA, tPA, PAI-1, uPA-PAI-1, tPA-PAI-1 (1 μg in 400 μl PBS), or PBS only (*n* = 5 per group; Fig 1A). In selected experiments, mice received neutrophil-depleting anti-Ly-6G or isotype control antibodies (*n* = 4 per group; Fig 1B). Subsequently, leukocyte-endothelial cell interactions were studied in postcapillary venules of the cremaster muscle of CX₃CR-1$^{+/GFP}$ mice by multi-channel *in vivo* microscopy 6 h after intrascrotal injection of recombinant murine uPA, PAI-1, uPA-PAI-1, TNF (1 μg in 400 μl PBS), or PBS only (*n* = 4 per group; Fig 1C). In further experiments, the effect of compound WX-340 (10 mg/kg body weight i.p.) on leukocyte-endothelial cell interactions was assessed in postcapillary venules of the cremaster muscle of CX₃CR-1$^{+/GFP}$ mice by multi-channel *in vivo* microscopy 6 h after intrascrotal injection of recombinant murine TNF (1 μg in 400 μl PBS; *n* = 6 per group; Fig 4B). Furthermore, the effect of WX-340 on neutrophil infiltration and tumor growth was assessed in an orthotopic model of 4T1 breast cancer in female BALB/c mice receiving WX-340 (10 mg/kg body weight i.p., daily) or drug vehicle *a priori* (starting on the same day as tumor cell injection) or therapeutically (starting on day 7 after tumor cell injection; *n* = 4–6 per group; Fig 4D). In separate experiments, animals received neutrophil- or cMO-depleting antibodies (*n* = 7 per group). In addition, the effect of WX-340 on neutrophil-endothelial cell interactions in tumors was analyzed in an heterotopic model of 4T1 breast cancer in female BALB/c mice receiving WX-340 (10 mg/kg body weight i.p., daily, starting on the same day as tumor cell injection) or drug vehicle (*n* = 4–6 per group; Fig 4E). Moreover, the effect of WX-340 on neutrophil infiltration and tumor metastasis was assessed in a tumor metastasis model of 4T1 breast cancer in female BALB/c mice receiving WX-340 (10 mg/kg body weight i.p., daily) or drug vehicle *a priori* (starting on the same day as tumor cell injection) or therapeutically (starting on day 7 after tumor cell injection; *n* = 4–6 per group; Fig 4F; Appendix Fig S4A).

## Peritonitis assay

Leukocyte recruitment to the peritoneal cavity was studied 6 h after induction of peritoneal inflammation. Mice were sacrificed, and their peritoneal cavity was washed with 10 ml of ice-cold saline. The total number of leukocytes in the peritoneal lavage fluid was measured with the ProCyte Hematology analyzer (IDEXX, Westbrook, Maine, USA).

Samples were then immunostained using antibodies (0.25 μg in 100 μl PBS) directed against CD45 (APC/Cy7, BioLegend, San Diego, California, USA), CD11b (FITC, BioLegend, San Diego, California, USA), Gr-1 (PE, BioLegend, San Diego, California, USA), CD115 (APC, BioLegend, San Diego, California, USA), F4/80 (eFluor 450, eBioscience/Thermo Fisher, San Diego, California, USA), CD19 (PerCP/Cy5.5, BioLegend, San Diego, California, USA), CD4 (Alexa Fluor 700, BioLegend, San Diego, California, USA), or CD8a (PE/Cy7, BioLegend, San Diego, California, USA) for 30 min on ice. After lysing erythrocytes (1:10, BD FACS Lysing solution, BD Bioscience) and two washing steps with PBS, samples were resuspended in 200 μl PBS.

## Mouse cremaster assay

The surgical preparation of the mouse cremaster muscle was performed as previously described by Baez with minor modifications (Baez, 1973). Briefly, the left femoral artery of anesthetized mice was cannulated in a retrograde manner in order to allow administration of substances including antibodies or inhibitors. In a next step, the right cremaster muscle was exposed through a ventral incision of the scrotum. The muscle was then opened ventrally and spread over a pedestal of a custom-made microscopy stage. After the epididymis and testicle were carefully detached from the cremaster muscle, they were placed back into the abdominal cavity. Throughout the surgical preparation and *in vivo* microscopy, the muscle was superfused with warm buffered saline.

## Orthotopic 4T1 tumor model

Tumor cells of the cell line 4T1 (at a concentration of $2 \times 10^5$ cells/20 μl) were injected into the left mammary fat pad of Balb/C mice, followed by daily treatment with WX-340 (10 mg/kg body weight in 50 μl saline; i.p.) or vehicle for 14 days. In separate experiments, WX-340 (10 mg/kg body weight in 50 μl saline; i.p.) was administered daily from day 7 until day 14 after tumor cell inoculation. Tumor size was morphometrically quantified on a daily basis using a digital caliper. Two weeks after application of tumor cells, tumor tissue and blood were harvested. The tumors were weighed and then homogenized in 15 ml saline. Of each sample, 100 μl were then immunostained with 0.25 μg anti-mouse antibodies directed against CD45 (APC/Cy7, BioLegend, San Diego, California, USA), CD11b (PerCp/Cy5.5, BioLegend, San Diego, California, USA), Ly-6G (PE, BioLegend, San Diego, California, USA) and F4/80 (eFluor 450, eBioscience/Thermo Fisher, San Diego, California, USA) as well as in selected experiments CD19 (APC, BioLegend, San Diego, California, USA), CD8a (PE/Cy7, BioLegend, San Diego, California, USA), and anti-CD4 (Alexa Fluor 700, BioLegend, San Diego, California, USA) for 30 min on ice. Erythrocytes were lysed using a lysing solution (1:10, BD FACS

Lysing solution, BD Bioscience). After two washing steps with PBS, samples were resuspended in 200 µl PBS and analyzed using multi-channel flow cytometry.

### Heterotopic 4T1 tumor model

4T1 tumor cells were injected into the left auricle of Balb/c mice at a concentration of $2 \times 10^5$ cells/20 µl, followed by daily treatment with WX-340 (10 mg/kg body weight in 50 µl saline; i.p.) or vehicle for 7 days. On day 3 or day 7 after tumor cell injection, mice ears were placed on a custom-made microscopy stage. Upon intradermal application of anti-Ly-6G PE (BioLegend, San Diego, California, USA; *via* the tail vein) mAB, *in vivo* microscopy analyses of neutrophil interactions in the tumor microvasculature were performed.

### 4T1 tumor metastasis model

Mice were inoculated with $2 \times 10^5$ 4T1 tumor cells *via* tail vein injection, followed by daily treatment with WX-340 (10 mg/kg body weight in 50 µl saline; i.p.) or vehicle for 14 days. In separate experiments, WX-340 (10 mg/kg body weight in 50 µl saline; i.p.) was administered daily from day 7 until day 14 after tumor cell inoculation. Mice were sacrificed on day 14 after tumor cell inoculation. Subsequently, blood and organs were harvested (and homogenized) for further multi-channel flow cytometry analysis (see below).

### Flow cytometry

Employing multi-channel flow cytometry (Gallios, Beckman Coulter Inc, Brea, California USA), myeloid leukocytes were identified by the expression of CD45 and CD11b. After exclusion of macrophages *via* high expression of F4/80, these cells were further divided into neutrophils (Gr-1$^{high}$ CD115$^-$), classical monocytes (Gr-1$^{high}$ CD115$^+$), and non-classical monocytes (Gr-1$^{low}$ CD115$^+$). In separate experiments, CD4$^+$ T lymphocytes (CD45$^+$ CD11b$^-$ CD4$^+$), CD8$^+$ T lymphocytes (CD45$^+$ CD11b$^-$ CD8a$^+$), and B lymphocytes (CD45$^+$ CD11b$^-$ CD19$^+$) were analyzed. In the tumor metastasis assay (see above), tumor cells were identified as Tomato$^+$ cells. All results were further processed by using the FlowJo Software (Treestar, Ashland, Oregon, USA).

### In vivo microscopy

*In vivo* microscopy was performed using an AxioTech-Vario 100 Microscope (Zeiss MicroImaging GmbH, Goettingen, Germany), equipped with a Colibri LED light source (Zeiss MicroImaging GmbH) for fluorescence epi-illumination microscopy. All microscopy videos were obtained with an AxioCam Hsm digital camera using a 40× water immersion lens (0.5 NA, Zeiss MicroImaging GmbH) and processed with the AxioVision 4.6 software (Zeiss MicroImaging GmbH). Video records were later analyzed by using the imaging software Fiji (Schindelin *et al*, 2012).

In the cremaster muscle of male CX$_3$CR-1$^{+/GFP}$ mice, neutrophils were identified as Ly-6G$^+$ CX$_3$CR-1$^-$ cells, classical monocytes as Ly-6G$^-$ CX$_3$CR-1$^{low}$ cells, and non-classical monocytes as Ly-6G$^-$ CX$_3$CR-1$^{high}$ cells upon intravenous injection of PE-labeled, non-

depleting monoclonal anti-Ly-6G antibodies. In the heterotopic tumor model, neutrophils were identified as Ly-6G$^+$ cells (anti-Ly-6G mAb, clone 1A8; 5 µg i.v.). Rolling leukocytes were defined as those moving slower than the associated blood flow and quantified for 30 s per venule. Firmly adherent cells were determined as those resting in the associated blood flow for > 30 s and related to the luminal surface per 100 µm vessel length.

### Immunostaining and confocal microscopy

Tumors were surgically removed from tumor-bearing mice and embedded in Tissue-Tek (Sakura, Alphen am Rhein, Netherlands). After storing the samples at −80°C, sections were cut at 20 µm using a cryostat (Thermo Fisher Scientific) and mounted onto glass slides (Thermo Fisher Scientific). Subsequently, sections were fixed with 4% formaldehyde (Microcos, Garching, Germany) for 10 min at RT, followed by washing the slides in PBS for 10 min. Blocking and permeabilization was achieved by incubating the slides in 2 % BSA (Sigma Aldrich, St. Louis, Missouri, USA) in PBS with 0.001 % Triton X-100 (Sigma Aldrich) for 1.5 h at room temperature. Finally, sections were incubated with anti-CD31/PECAM-1 AF647 (BioLegend) and primary anti-uPA (Santa Cruz Biotechnology, Santa Cruz, CA/USA) or anti-PAI-1 antibody (Abcam, Cambridge, UK; for 4 h at room temperature; 1 µg in 200 µl PBS), before incubation with secondary Alexa Fluor 488-linked antibodies (Invitrogen, Carlsbad, CA; 1 µg in 200 µl PBS) in blocking solution at 4°C over night. After washing the slides twice in PBS for 5 min, samples were mounted using PermaFluor (Beckman Coulter, Brea, California, USA) and stored at 4°C.

In order to evaluate the expression of uPA or PAI-1 (6 h after intrascrotal injection of TNF) or ICAM-1/CD54 and VCAM-1/CD106 (6 h after intrascrotal injection of recombinant murine uPA, PAI-1, uPA-PAI-1 or saline) in cremasteric tissue, excised mouse cremaster muscles were fixed in 4% paraformaldehyde. Next, tissues were blocked and permeabilized using 2% BSA in PBS with 0.001% Triton X-100 for 1.5 h at room temperature. Sections were then incubated with anti-CD31/PECAM-1 (AF647; BioLegend) and primary anti-uPA (Santa Cruz Biotechnology, Santa Cruz, CA/USA) or anti-PAI-1 antibodies (Abcam, Cambridge, UK; for 4 h at room temperature; 1 µg in 200 µl PBS), before incubation with secondary Alexa Fluor 488-linked antibodies (Invitrogen, Carlsbad, CA; 1 µg in 200 µl PBS) in blocking solution at 4°C over night. In separate experiments, sections were incubated with anti-CD31/PECAM-1 (AF647; BioLegend) as well as with anti-ICAM-1/CD54 (Alexa Fluor 488, BioLegend) or anti-VCAM-1/CD106 antibodies (APC, BioLegend). Finally, immunostained sections were mounted in Perma-Fluor (Thermo Fisher Scientific) on glass slides. Confocal z-stacks (z-spacing 1 µm) were acquired using a Leica SP8 confocal laser scanning microscope (Leica Microsystems, Wetzlar, Germany) with an oil-immersion lens (Leica; 40×; NA 1.40). The fluorescence signal was quantified using the software Fiji. Background signal was subtracted.

### Spinning disk confocal microscopy in autoperfused flow chamber assays

CD11a/LFA-1 clustering under flow was evaluated by time-lapse microscopy using an upright spinning disk confocal microscope

(Examiner; Zeiss) with a confocal scanner unit CSU-X1 (Yokogawa Electric Corporation, Japan), an EMCCD camera (Evolve; Photometrics), and a x20/1.0 NA water immersion objective (Plan Apochromat; Zeiss). For flow chamber experiments, IBIDI-Slide IV 0.1 flow chambers (Ibidi, Munich, Germany) were coated for 3 h with a combination of 20 μg/ml CD62E/E-Selectin and 15 μg/ml CD54/ICAM-1 (Fc chimera, R&D Systems), 15 μg/ml CXCL1 (Peprotech), and/or 15 μg/ml uPA-PAI-1 heteromers (Molecular Innovations). For analysis of CD11a/LFA-1 clustering, neutrophils from murine bone marrow were isolated with Percoll (Density 1.08/1.11, Sigma) and incubated overnight with WEHI supernatant and 1 μl SIR-actin (F-actin Labeling Probe, Spirochrome). Subsequently, neutrophils were incubated for 10 min with a non-blocking rat anti-mouse CD11a/LFA-1 Alexa Fluor 546 antibody (clone 2D7, BioLegend) and a neutrophil-specific rat anti-mouse Ly6G Alexa Fluor 488 antibody (Clone 1A8, BioLegend). Flow chambers were then filled with labeled neutrophils and allowed to attach for 5 min. Thereafter, chambers were flushed, and detachment assays performed over 10 min with increasing flow rates (0.5–40 dyne/cm$^2$) using high-precision syringe pump (Model KDS-232, KD Scientific, USA) and recorded as time-lapse movie. Movies were then analyzed offline for CD11a/LFA-1 and actin colocalization (signal intensity) using Fiji software.

### Depletion of neutrophils or classical monocytes

For the depletion of circulating neutrophils, mice received injections of anti-Ly-6G monoclonal antibodies (clone 1A8; 50 μg intravenously (i.v.); 24 h and 6 h prior to induction of inflammation in the peritonitis model or every other day for one week in the tumor model; BD Biosciences, San Jose, CA, USA) as described elsewhere (Zuchtriegel *et al* 2016). This specifically depletes neutrophils (Daley *et al* 20082008) by Fc-dependent opsonization and phagocytosis of the antibody-bound cells (Bruhn *et al* 2016). For depletion of classical monocytes in the tumor model, mice received injections of anti-CCR2 antibodies (clone MC-21; 25 μg i.v. every day for one week; provided by Matthias Mack, Department of Internal Medicine, University of Regensburg, Regensburg, Germany; Bruhl *et al*, 2007).

### ELISA analysis of uPA-PAI-1 heteromerization

The effect of WX-340 on heteromerization of uPA and PAI-1 was analyzed using an enzyme-linked Immunosorbent Assay (ELISA). First, wells of a 96-well plate were coated with recombinant murine PAI-1 (Molecular Innovations, Novi, Michigan, USA) at 4°C overnight. After washing three times with PBS, the wells were incubated with recombinant uPA (0, 1, 5, 10 ng/ml, Molecular Innovations, Novi, Michigan, USA) each combined with different doses of the inhibitor WX-340 (0, 0.1, 1, 10 μM). After incubation for 60 min and subsequent washing of the wells three times with PBS, the amount of uPA bound to PAI-1 in the wells was quantified employing a Mouse uPA total antigen assay ELISA kit (Molecular Innovations, Novi, Michigan, USA).

### Multiplex cytokine ELISA analyses

Cytokine production by RAW 264.7 macrophages stimulated with recombinant murine uPA-PAI-1 (1 μg/ml) was analyzed in undiluted culture supernatants using the Bio-Plex Pro™ Mouse chemokine panel 33-Plex on a Bio-Plex 200 system according to the manufacturer's protocol (Bio-Rad laboratories, Munich, Germany). Group comparisons were performed by unpaired Student's *t*-test with subsequent Benjamini–Hochberg correction. FDR *q*-value < 0.1 was used as cutoff for statistical significance.

### Activation of neutrophils

As a measure of neutrophil activation, surface expression of the integrins LFA-1/CD11a, Mac-1/CD11b, and VLA-4/CD49d was determined in anticoagulated blood samples incubated for 30 min with recombinant murine uPA-PAI-1 (100 ng/ml) or saline. Subsequently, samples were washed with PBS and cells were immunostained using antibodies (0.25 μg in 100 μl PBS) directed against CD45 (APC/Cy7, BioLegend, San Diego, California, USA), CD11b (PerCp/Cy5.5, BioLegend, San Diego, California, USA), Ly-6G (PE, BioLegend, San Diego, California, USA) and F4/80 (eFluor 450, eBioscience/Thermo Fisher, San Diego, California, USA), and CD11a (PE/Cy7, eBioscience), and CD49d (FITC, eBioscience). Lyses of erythrocytes with lysing solution followed. After washing the samples twice in PBS, samples were resuspended in 200 μl PBS and analyzed by multi-channel flow cytometry.

As a measure of conformational changes of integrins, binding of ICAM-1/CD54-Fc to neutrophils was analyzed. Briefly, blood was taken from WT mice, anticoagulated, and suspended in Hanks balanced salt solution containing 1 mM $CaCl_2$ and $MgCl_2$ (Life Technologies, Carlsbad, California, USA). Subsequently, cells were incubated with recombinant murine uPA-PAI-1 (1 μg/ml) or PBS as negative control for 30 min at 37°C, followed by adding ICAM-1/CD54-Fc (10 μg/ml, R&D Systems) and PE-conjugated anti-human IgG1 (10 μg/ml, Fc-specific, Southern Biotechnology, Birmingham, Alabama, USA) for 5 min at 37°C. Next, cells were labeled with antibodies (0.25 μg in 100 μl PBS) directed against CD45 (APC/Cy7, BioLegend, San Diego, California, USA), CD11b (FITC, BioLegend, San Diego, California, USA), Ly-6G (Alexa Fluor 700, BioLegend, San Diego, California, USA), and F4/80 (eFluor 450, eBioscience/Thermo Fisher, San Diego, California, USA). Binding of ICAM-1/CD54-Fc to neutrophils was measured by using multi-channel flow cytometry.

The conformation-specific antibodies (0.25 μg in 100 μl PBS) mAb24 (high-affinity conformation of β2 integrins; mouse anti-human, monoclonal; Abcam, Cambridge, United Kingdom) and kim127 (intermediate-affinity and high-affinity conformation of β2 integrins; mouse anti-human, monoclonal) were used to analyze the integrin conformation status in human neutrophils after incubation with uPA-PAI-1 recombinant murine uPA-PAI-1, TNF (100 ng/ml), or saline for 30 min by multi-channel flow cytometry.

### Activation of endothelial cells

To measure activation of cultured mouse endothelial cells (bEnd.3), cells were seeded into 12-well plates and exposed to recombinant murine uPA-PAI-1, TNF (100 ng/ml), or saline for 6 h at 37°C. Cells were collected and resuspended in saline before they were then immunostained using antibodies (0.25 μg in 100 μl PBS) directed against ICAM-1/CD54 (Alexa Fluor 488, BioLegend), VCAM-1/CD106 (APC, BioLegend), primary anti-uPA (Santa Cruz Biotechnology, Santa Cruz, CA/USA), or primary anti-PAI-1 antibodies (Abcam,

Cambridge, UK) followed by incubation with secondary Alexa Fluor 488-linked antibodies (Invitrogen, Carlsbad, CA). In separate experiments, bEnd.3 endothelial cells were co-cultured for 6 h with RAW macrophages, which were, prior to this, exposed for 3 h to recombinant mouse TNF, uPA-PAI-1 (100 ng/ml), or saline. After washing the samples twice in PBS, samples were resuspended in 200 µl PBS and analyzed by multi-channel flow cytometry.

### TNF production in macrophages

To measure intracellular TNF production of cultured mouse macrophages (RAW 264.7) or peritoneal macrophages from C57BL/6J mice, cells were seeded into 12-well plates, treated with Brefeldin A und Monensin (Protein Transport Inhibitor Cocktail, eBioscience/ Thermo Fisher, San Diego, California, USA) and then exposed to recombinant murine uPA-PAI-1 (1 µg/ml) or saline for 6 h at 37°C. After detaching and resuspending the cells in saline, intracellular TNF was then detected by using antibodies directed against TNF (0.25 µg in 100 µl PBS; PE, BioLegend, San Diego, California, USA) after permeabilizing cells with an intracellular staining permeabilization wash buffer (BioLegend, San Diego, California, USA). After washing the samples twice in PBS, samples were resuspended in 200 µl PBS and analyzed by multi-channel flow cytometry.

### Cell proliferation assay

4T1 tumor cells or bEnd.3 endothelial cells were seeded on 96-well plate and exposed to WX-340 (10 µM), recombinant murine uPA-PAI-1 (1 µg/ml), or vehicle. After 48 h, serum-free media and the MTT reagent were added according to the manufacturer's protocol (Abcam, Cambridge, UK) for 3 h at 37°C. Next, the MTT solvent was added and the plate was placed on an orbital shaker for 15 min. Finally, absorbance was measured at 590 nm in a microplate reader (Tecan, Männedorf, Switzerland). Cell proliferation was determined as the percentage of change as compared to the negative control after background subtractions.

In separate experiments, neutrophils were isolated with the EasySepTM Mouse Neutrophil Enrichment Kit (STEMCELL Technologies, Vancouver, Canada) from the peritoneal cavity of C57BL/6J mice 6 h after intraperitoneal injection of recombinant murine uPA-PAI-1 (1 µg/ml), as described by the manufacturer. Isolated neutrophils were placed in the cell culture and incubated over night at 37°C. On the following day, supernatants from the isolated neutrophils were placed onto tumor cells or endothelial cells and exposed to the NE inhibitor and/or WX-340. After incubation for 24 h at 37°C, the MTT assay was performed.

### Human breast cancer samples

Written informed consent was obtained from all subjects. The experiments conformed to the principles set out in the WMA Declaration of Helsinki and the Department of Health and Human Services Belmont report. The protocol was approved by the ethics committee of the Heidelberg University Clinical Centre, Heidelberg/Germany.

An overall of 44 breast cancer cases, which were surgically treated in the University Hospital Heidelberg, Germany, were included in the cohort. Cohort details are given in Appendix Table S1. In all these cases, the commercially available diagnostic grade ELISA-based

### The paper explained

#### Problem

Breast cancer is the most common oncological disorder in women worldwide. High intratumoral levels of heteromers of the serine protease urokinase-type plasminogen activator (uPA) and its inhibitor plasminogen activator inhibitor-1 (PAI-1) predict impaired survival and treatment response already in early stages of breast cancer. Although these single proteins are well known to control tissue perfusion by regulating clot formation as key components of the fibrinolytic system, the pathogenetic role of this protein complex in breast cancer remains obscure.

#### Results

Utilizing patient data and different syngeneic mouse models of breast cancer, we demonstrate that heteromerization of uPA and PAI-1 multiplies the potential of the single proteins to attract pro-tumorigenic neutrophils. To this end, tumor-released uPA-PAI-1 utilizes the very low-density lipoprotein receptor and intracellular mitogen-activated protein kinases to initiate a pro-inflammatory program in perivascular macrophages in the proximity of malignant tumors. This enforces neutrophil trafficking to cancerous lesions and skews these immune cells toward a pro-tumorigenic phenotype, thus supporting tumor growth and metastasis. Blockade of uPA-PAI-1 heteromerization by a novel small-molecule inhibitor interfered with these events and effectively prevented tumor progression.

#### Impact

Our findings identify a therapeutically targetable, hitherto unknown interplay between hemostasis and innate immunity that drives breast cancer progression. As a personalized immunotherapeutic strategy, blockade of uPA-PAI-1 heteromerization might be particularly beneficial for patients with highly aggressive uPA-PAI-1[high] tumors.

FEMTELLE® uPA/PAI-1 assay (https://www.femtelle.de/de/) was performed on the obtained surgical specimens in the context of routine diagnostic workup. PAI-1 as well as uPA protein concentrations were extracted from the respective reports.

For the exact quantification of neutrophils in the same tumor cohort, slides were cut from formalin-fixed paraffin-embedded tissue blocks comprising the central tumor area. Immunohistochemistry was performed on a BenchMark XT automated stainer (Ventana, Tucson, AZ) with an antibody directed against myeloperoxidase (Thermo Fisher Scientific, RB-373-A) using the ultraVIEW DAB Detection Kit (all reagents from Ventana, Tucson, AZ). Briefly, the tissue sections were deparaffinized with EZ Prep at 75°C and 76°C, heat pretreated in Cell Conditioning 1 (CC1) for antigen retrieval at 76°C–100°C, and then incubated with the primary antibody diluted in antibody diluent 1:100 for 28 min at 37°C after inactivation of the endogenous peroxidase using UV inhibitor for 4 min at 37°C. Antibody binding was detected using DAB as chromogen and counterstained with hematoxylin for 10 min with subsequent bluing in bluing reagent for 10 min. Slides were then dehydrated manually by alcohol washes of increasing concentrations (70%, 96%, 100%) and xylene and cover-slipped using Pertex® mounting medium (Histolab, 00801).

Stained slides were scanned with a high-throughput slide scanner (AT2, Leica Microsystems), and tumor areas were annotated by a board-certified pathologist (WW) using a Wacom Cintiq22HD display (Wacom) on the eSlideManager database (Leica Biosystems). Neutrophils were quantified by counting stained cells in the core tumor region pre-marked by the pathologist in a blinded manner and normalized to mm$^2$.

## Analyses of RNA microarray data from the METABRIC breast cancer cohort

For analyses of RNA microarray data of the METABRIC breast cancer cohort (Curtis *et al*, 2012), CBioportal was employed (Gao *et al*, 2013). The subcohort of patients with documented tumor stage and available RNA microarray data was used ($n = 1,466$), and follow-up was cut to 120 months. RNA co-expression analyses were done utilizing Spearman and Pearson algorithms. Survival analyses were performed in subgroups of different tumor stages with the Cox proportional hazard model, and target gene expression z-values of $\geq 2.0$ were used as strata.

## Statistics

Data analysis was performed using the statistical software package (SigmaStat for Windows; Jandel Scientific). Unpaired Student's *t*-tests (2 groups) or one-way ANOVA test followed by multiple *post hoc* tests with Tukey's correction (> 2 groups) was used for the estimation of stochastic probability in intergroup comparisons. Data are presented as mean ± SEM. *P*-values <0.05 were considered significant.

# Data availability

This study includes no data deposited in external repositories.

**Expanded View** for this article is available online.

## Acknowledgments

This study is part of the doctoral thesis of J.C.D. We thank the NCT Tissue Bank in Heidelberg, Germany, and the MTBIO biobank as well as the Comparative Experimental Pathology (CEP) Unit at the TU Munich, Munich, Germany, for technical support. This study was supported by Deutsche Forschungsgemeinschaft (DFG, SFB 914, projects B01 to M.S., B03 to C.A.R., B06 to K.L., Z03 to M.S.). In addition, certain aspects of the study were supported by a grant of the German Cancer Aid to W.W. as part of the INTEGRATE-TN consortium. Open Access funding enabled and organized by Projekt DEAL.

## Author contributions

BU and CAR conceived the study and wrote the manuscript. BU, LM, JD, RH, BS, FH, JS, CB, LP, RP, and KS performed experiments and data analysis. WW, MS, KL, and CAR contributed to data analysis. MC, CS, MM, EG, PS, JH, SMK, WW, MS, KL, and FK contributed to the writing of the manuscript.

## Conflict of interest

The authors declare that they have no conflict of interest.

## For more information

https://www.femtelle.de

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
