## [Review Process File · EMBO Molecular Medicine]

uPA-PAI-1 heteromerization promotes breast cancer progression by attracting tumorigenic neutrophils

Bernd Uhl, Laura Mittmann, Julian Dominik, Roman Hennel, Bojan Smiljanov, Florian Haring, Johanna Schaubaecher, Constanze Braun, Lena Padovan, Mathias Mack, Robert Pick, Martin Canis, Sandip Kanse, Christian Schulz, Ewgenija Gutjahr, Hans-Peter Sinn, Jörg Heil, Katja Steiger, Wilko Weichert, Markus Sperandio, Kirsten Lauber, Fritz Krombach, and Christoph Reichel

DOI: [10.15252/emmm.202013110](https://doi.org/10.15252/emmm.202013110)

Corresponding author: Christoph Reichel (christoph.reichel@med.uni-muenchen.de)

Review Timeline:

Submission Date:	14th Jul 20
Editorial Decision:	24th Aug 20
Revision Received:	22nd Feb 21
Editorial Decision:	12th Mar 21
Revision Received:	22nd Mar 21
Accepted:	25th Mar 21

Editor: Lise Roth

Transaction Report:

24th Aug 2020

Dear Prof. Reichel,

Thank you for the submission of your manuscript to EMBO Molecular Medicine, and please accept my apologies for the delay in getting back to you. We have now received feedback from two of the three reviewers who agreed to evaluate your manuscript. Given that referee #1 has unfortunately not returned his/her report so far despite several chasers, and that both referees 2 and 3 are overall positive, we prefer to make a decision now in order to avoid further delay in the process. Should referee #1 provide a report, we will send it to you, with the understanding that we would not ask you for extensive experiments in addition to the ones required in the enclosed reports.

As you will see from the reports below, both referees acknowledge the interest of the study and are overall supporting publication of your work pending appropriate revisions.

Addressing the reviewers' concerns in full will be necessary for further considering the manuscript in our journal, and acceptance of the manuscript will entail a second round of review. EMBO Molecular Medicine encourages a single round of revision only and therefore, acceptance or rejection of the manuscript will depend on the completeness of your responses included in the next, final version of the manuscript. For this reason, and to save you from any frustrations in the end, I would strongly advise against returning an incomplete revision.

When submitting your revised manuscript, please carefully review the instructions that follow below. Failure to include requested items will delay the evaluation of your revision:

- 1) A .docx formatted version of the manuscript text (including legends for main figures, EV figures and tables). Please make sure that the changes are highlighted to be clearly visible.
- 2) Individual production quality figure files as .eps, .tif, .jpg (one file per figure).
- 3) A .docx formatted letter INCLUDING the reviewers' reports and your detailed point-by-point responses to their comments. As part of the EMBO Press transparent editorial process, the point-by-point response is part of the Review Process File (RPF), which will be published alongside your paper.
- 4) A complete author checklist, which you can download from our author guidelines (<https://www.embopress.org/page/journal/17574684/authorguide#submissionofrevisions>). Please insert information in the checklist that is also reflected in the manuscript. The completed author checklist will also be part of the RPF.
- 5) Please note that all corresponding authors are required to supply an ORCID ID for their name upon submission of a revised manuscript.
- 6) Before submitting your revision, primary datasets produced in this study need to be deposited in an appropriate public database (see <https://www.embopress.org/page/journal/17574684/authorguide#dataavailability>). Please remember to provide a reviewer password if the datasets are not yet public.

The accession numbers and database should be listed in a formal "Data Availability " section (placed after Materials & Method). Please note that the Data Availability Section is restricted to new primary data that are part of this study.

7) We would also encourage you to include the source data for figure panels that show essential data. Numerical data should be provided as individual .xls or .csv files (including a tab describing the data). For blots or microscopy, uncropped images should be submitted (using a zip archive if multiple images need to be supplied for one panel). Additional information on source data and instruction on how to label the files are available at .

8) Our journal encourages inclusion of *data citations in the reference list* to directly cite datasets that were re-used and obtained from public databases. Data citations in the article text are distinct from normal bibliographical citations and should directly link to the database records from which the data can be accessed. In the main text, data citations are formatted as follows: "Data ref: Smith et al, 2001" or "Data ref: NCBI Sequence Read Archive PRJNA342805, 2017". In the Reference list, data citations must be labeled with "[DATASET]". A data reference must provide the database name, accession number/identifiers and a resolvable link to the landing page from which the data can be accessed at the end of the reference. Further instructions are available at .

9) We replaced Supplementary Information with Expanded View (EV) Figures and Tables that are collapsible/expandable online. A maximum of 5 EV Figures can be typeset. EV Figures should be cited as 'Figure EV1, Figure EV2" etc... in the text and their respective legends should be included in the main text after the legends of regular figures.

- Additional Tables/Datasets should be labeled and referred to as Table EV1, Dataset EV1, etc. Legends have to be provided in a separate tab in case of .xls files. Alternatively, the legend can be supplied as a separate text file (README) and zipped together with the Table/Dataset file. See detailed instructions here: .

10) Every published paper now includes a 'Synopsis' to further enhance discoverability. Synopses are displayed on the journal webpage and are freely accessible to all readers. They include a short stand first (maximum of 300 characters, including space) as well as 2-5 one-sentences bullet points that summarizes the paper. Please write the bullet points to summarize the key NEW findings. They should be designed to be complementary to the abstract - i.e. not repeat the same text. We encourage inclusion of key acronyms and quantitative information (maximum of 30 words / bullet point). Please use the passive voice. Please attach these in a separate file or send them by email, we will incorporate them accordingly.

Please also suggest a striking image or visual abstract to illustrate your article. If you do please provide a png file 550 px-wide x 400-px high.

11) As part of the EMBO Publications transparent editorial process initiative (see our Editorial at <http://embomolmed.embopress.org/content/2/9/329>), EMBO Molecular Medicine will publish online a Review Process File (RPF) to accompany accepted manuscripts.

In the event of acceptance, this file will be published in conjunction with your paper and will include the anonymous referee reports, your point-by-point response and all pertinent correspondence relating to the manuscript. Let us know whether you agree with the publication of the RPF and as here, if you want to remove or not any figures from it prior to publication.

I look forward to receiving your revised manuscript.

Yours sincerely,

Lise Roth

Lise Roth, PhD
Editor
EMBO Molecular Medicine

To submit your manuscript, please follow this link:

Link Not Available

Photos 400-800 DPI

*Additional important information regarding figures and illustrations can be found at <http://bit.ly/EMBOPressFigurePreparationGuideline>

***** Reviewer's comments *****

Referee #2 (Comments on Novelty/Model System for Author):

The technical quality of the present manuscript is high and the employed model systems are adequate. The novelty of the described findings are medium as uPA-PAI1 has already been associated with neutrophil trafficking by the authors elsewhere and effects of WX-340 on breast cancer progression have already been described by others. An application of WX-340 in therapy needs lengthy approval thus a direct medical impact is not given.

Referee #2 (Remarks for Author):

The authors of the manuscript "uPA-PAI-1 heteromerization promotes breast cancer progression by attracting tumorigenic neutrophils" show that tumors release urokinase-type plasminogen activator (uPA)-plasminogen activator inhibitor-1 (PAI-1) heteromers and initiate a pro-inflammatory program in perivascular macrophages leading to the attraction of pro-tumorigenic neutrophils. Inhibition of uPA-PAI-1 heteromerization by WX-340 decreased neutrophil trafficking and had prophylactic and therapeutic effects in vivo.

The manuscript is well written and technically sound.

Certain effects such as the effect of uPA-PAI1 on neutrophil trafficking especially compared to single proteins are rather weak and the depicted assays (Figure 1c) lack positive controls. The authors might want to critically discuss this.

It would be interesting to compare the effect of uPA-PAI1 to single proteins alone in further assay such as the cremaster expression of ICAM and VCAM after injection.

The prophylactic effect of the inhibitor could be tested in a model of induced breast cancer to underline its efficacy and applicability.

Some typos such as microvascular (page6) and their (which should be "the" on page 7)...

Referee #3 (Comments on Novelty/Model System for Author):

The findings are interesting but it very technical without clear explanation for general readership. It need expanding of the text especially the results for this to becomes more clear to a reader who does not have immunological expertise or imaging of the particular technology used by the authors.

Referee #3 (Remarks for Author):

In this manuscript, Uhl et al. present evidence for the role of urokinase-type plasminogen activator (uPA)-plasminogen activator inhibitor-1 (PAI-1) heteromers in breast cancer. The authors demonstrate the potential for clinical relevance by associating uPA and PAI-1 protein levels with both extravascular and intravascular neutrophils, from breast cancer cohort patients, that were diagnosed with G1 breast cancer. uPA-PAI-1 heteromers are part of the fibrinolytic system, impacting cell adhesion and migration, angiogenesis, signal transduction and apoptosis. uPA-PAI-1

can recruit both macrophages and neutrophils and tumor derived uPA-PAI-1 protein expression, is known to impact breast cancer and influence metastasis. The authors utilized a mouse peritoneal assay(s), along with multi-channel flow cytometry, to track neutrophil motility in response to uPA-PAI-1 heteromers treatment. In response to uPA-PAI-1 exposure neutrophils, increased transmigration activity, in the intertumoral space thus promoting early tumor growth and metastasis. In vivo, uPA-PAI-1 exposure increases the number of both macrophage and neutrophils into the peritoneal cavity, neither B and or T lymphocytes were similarly recruited after treatment. The activity of uPA-PAI-1 does not depend on activating endothelial cells expression of either ICAM and or VCAM, for transmigration activity. Rather uPA-PAI-1 heteromers activity increase ICAM and VCAM expression in a neutrophil specific manner through the supposed activity of TNF. The authors present data that increase TNF production arises from the macrophages in response to uPA-PAI-1 exposure. This would suggest that uPA-PAI activation is upstream of TNF as it increases ICAM and VCAM expression in neutrophils but not in the endothelial cells. Moreover, uPA-PAI-1 activity in vivo, that influenced the NE neutrophils subtype rather than MMP9 and or VEGF subtype. The NE neutrophils stimulate 4T1 proliferation in vitro and in vivo uPA-PAI-1 inhibitor (WX340) decreased neutrophil recruitment delaying 4T1 cells tumor growth and decreasing metastasis.

General comments:

In general, the manuscript has very intriguing results and mostly well control. Description(s) of the experimental methodology was often lacking in details within the text causing one to pause and ask was the specific assay been conducted in vivo or in vitro? Description of muscle cremaster models using the Ly-6G+ CX3CR-1- cells is poorly described in this manuscript it is referenced as published, should be introduce for general readership. Some results where conflated in one experimental system to conclude similar findings in a second experimental system and suggesting the impact on cellular activity was the same without providing raw data for independent assessment of these conclusions. Tumors biology lack tracking of tumor growth curves either daily or weekly. Histomorphogenic analysis of tumors and metastasis lesion was absent to ensure the arguments that this was due to less neutrophil recruitment. Validation of the IHC analysis to make conclusion in regards blocking growth and or preventing metastasis was not convincing. In all the manuscript has made a very important observation(s) and conclusions that merit considerations for publication. This would require major editing and response to several major and minor points so that impact of the work is broadly accepted.

Major Specific points:

1. Fig. 1B it is not clear why the author depleted the neutrophils with Ly-6B yet identify the neutrophils as Cd11bhi/GR-1hi/CD115 neg, the raw FACS data needs to be shown. Ly-6B is a general granulocyte marker also found in eosinophils and basophils. Ly-6B is also found in activate macrophages and uPA-PAI-1 activated macrophage recruitment to the peritoneal cavity. So it is misleading to called it a "neutrophil depletion" when it also depletes macrophages as shown by the figure 1A.
2. The neutralizing assay is not detailed in M&M and which antibody was used to do the depletion.
3. Fig 1C As this model is heavily dependent on many of its I conclusion on microcopy imaging it is quite unclear why the cells are label Green (GR1 positive) when utilizing the Ly-6G+ GFP- using the Cx3CR-1GFP+ mice. This highlight the difficult the author creates by not describing their model for this particular manuscript with little details and relying on previous published work to call the cells neutrophils. This manuscript should stand alone without having to go read another manuscript.
4. A single image is not representative of the quantified data shown for adherence and transmigration assays for either the macrophages and or the neutrophils the data is not impressive showing two cells.
5. Fig 2a. Cremasteric assay no reference provide in text for this manuscript. Immunofluorescent green for UPA and PAI are they labels with green tags, is unclear. Upon TNF treatment recruitment

- of neutrophils are pink and green the protein moved it is just not clear at all or are the uPA-PAI-1 heterodimerization occurring in the neutrophils or another cell subtype, this is not at all clear.
6. Fig 2C an in vitro assay with RAW 264.7 exposed to uPA-PAI-1 heteromers is not the same thing as macrophages derived from an in vivo source. Especially if the arguments are that macrophages are been recruit to assist the neutrophils too attach and trans migrate. As they provide the chemokines and our TNF production as the source to elicit such action in vivo, that is trans migration.
 7. Fig2D results using an endothelial in vitro assay vs what was shown in Fig2B in vivo are not consistent, in one assay the uPA-PAI-1 heteromers increase ICAM/VCAM but in the other I assay it does not? If the author were attempting to imply that the in vivo assay requires macrophage recruitment for uPA-PAI-1 to the endothelial cell for both ICAM/VCAM to be up regulated only in vivo but not in vitro this was not clear at all in the text or the explanations of the results. Therefore, culture of the bEND.3 with the RAW cells would potentially address this issue when exposed to either TNF and or uPA-PAI-1 heteromers.
 8. Fig1E require images for independent assessment that cell is specifically interacting with one cell surface receptor but not the other.
 9. Fig EV4 needs to show IHC results of the neutrophil population in G1, G2 and G3 breast cancer samples. For independent assessment of authors conclusions.
 10. Figure 3A a single IHC for uPA-PAI-1 is not valid for this analysis and will require showing of multiple samples with wider range of the tumor showing.
 11. The author do not address the expression for uPA-PAI-1 has been shown that can also come from human tumors and not just the infiltrating neutrophils.
 12. Figure 3b the cohort analyzed for the METABRIC data set need to be clarify as to which subtype of breast cancer were analyzed. Where there any differences between Luminal A and or B, Her 2 and or basal/TNBC subtypes. No p-value and or HR are shown diminishing the significance of the study. Simple correlation are not indicate of clinical relevance.
 13. Figure 3d it is not clear why the inhibitor WX-340 was not added to the neutrophils plus 4t1 cells or when NE inhibitor was added to show combinatorial effects.
 14. Fig4a not clear what cells are been used either the text or the figure legends. Similar for Fig4b not clear if this is in the context of the tumor in vivo model or not.
 15. Fig4C the staining for uPA and or PAI-1 are unimpressive and do not appear to be specific but rather a wide expression pattern is shown to most every cell.
 16. The author does not rule out the effects on lack of macrophage recruitment to the tumor biology his needs to be addressed.
 17. Fig4D tumor growth kinetics are lacking along with histomorphogenic analysis of tumors.
 18. Figure 4F effects on metastasis by 4T1 cells is very robust to multiple organs yet these results cannot be independently asses by the data shown.
 19. IHC and HE comparison are requires and timing of the experiment metastasis is also lacking.

Referee #2

General comment: The technical quality of the present manuscript is high and the employed model systems are adequate. The novelty of the described findings are medium as uPA-PAI-1 has already been associated with neutrophil trafficking by the authors elsewhere and effects of WX-340 on breast cancer progression have already been described by others. An application of WX-340 in therapy needs lengthy approval thus a direct medical impact is not given.

The authors of the manuscript "uPA-PAI-1 heteromerization promotes breast cancer progression by attracting tumorigenic neutrophils" show that tumors release urokinase-type plasminogen activator (uPA)-plasminogen activator inhibitor-1 (PAI-1) heteromers and initiate a pro-inflammatory program in perivascular macrophages leading to the attraction of pro-tumorigenic neutrophils. Inhibition of uPA-PAI-1 heteromerization by WX-340 decreased neutrophil trafficking and had prophylactic and therapeutic effects in vivo. The manuscript is well written and technically sound.

Answer: We appreciate the very positive evaluation of our manuscript. Indeed, we and others previously investigated the role of uPA, PAI-1, and other components of the fibrinolytic system for neutrophil migration to sites of inflammation. The present study, however, deciphers in detail a distinct role of the heteromerization process of uPA and PAI-1 for neutrophil trafficking which multiplies the action of the single proteins – in striking contrast to the critical inhibitory effect of uPA-PAI-1 heteromerization on fibrinolysis.

We are more than happy to cite previous findings on the effect of compound WX-340 in breast cancer progression. To our knowledge, however, published experimental data on effects of WX-340 are limited to models of amyotrophic lateral sclerosis (Glas et al., *Exp Neurol* 2007), rheumatoid arthritis (Serrati et al., *Arthritis Rheum* 2011), and ischemia-reperfusion injury (Reichel et al., *Circulation* 2011).

Comment 1.) Certain effects such as the effect of uPA-PAI-1 on neutrophil trafficking especially compared to single proteins are rather weak and the depicted assays (Figure 1c) lack positive controls. The authors might want to critically discuss this.

Answer: Thank you very much for this comment. We now included a positive control in **Appendix Fig. S1a** using the well-known cytokine tumor necrosis factor (TNF) as an inflammatory stimulus (1 µg; same dose as uPA-PAI-1 in our experiments). Our data clearly demonstrate similar effects of TNF and uPA-PAI-1 heteromers on neutrophil trafficking in our model system. This point is now discussed in the revised version of our manuscript (p. 6, 1st paragraph; p. 13, 1st paragraph).

Comment 2.) It would be interesting to compare the effect of uPA-PAI-1 to single proteins alone in further assay such as the cremaster expression of ICAM and VCAM after injection.

Answer: According to the reviewer's suggestion, we now show that the effect of uPA-PAI-1 heteromers on the endothelial expression of ICAM-1/CD54 and VCAM-1/CD106 is more pronounced as compared to the single proteins. These novel data are included and discussed in the revised version of our manuscript (**Appendix Fig. S1b**; p. 6, 2nd paragraph; p. 13, 2nd paragraph).

Comment 3.) The prophylactic effect of the inhibitor could be tested in a model of induced breast cancer to underline its efficacy and applicability.

Answer: We appreciate the reviewer's comment that the effect of WX-340 could be evaluated in a model of induced breast cancer. However, exploring immune cell responses in induced breast cancer models is associated with significant limitations since the established modes of tumor induction themselves modulate immune cell responses and thereby might interfere with tumor-elicited immune cell trafficking:

Estradiol is used in **hormonal** breast cancer induction which is well known to change the production of pro-inflammatory factors (Vieira et al., Transplantation 2020), leukocyte responsiveness (Buyon et al., Arthritis Rheum 1984), and leukocyte survival (Otonello et al., Ann N Y Acad Sci 2002), thus affecting immune cell trafficking. Similarly, **chemical** substances employed for experimental breast cancer induction such as DMBA (1,3-Dimethylbenz(a)anthracene), NMU (N-methyl-N-nitrosourea), PhiP (2-Amino-1-methyl-6-phenylimidazo[4,5-B]pyridine), or MC (Methylcholanthrene) exhibit potent myelotoxic effects (DMBA: Katz et al., Int J Toxicol 2014), modulate the production of inflammatory mediators (NMU: Li et al., Canc Res 2002; PhiP: Yun et al., Toxicology 2006), and induce apoptosis in leukocytes (MC: Reynaud et al., Aquat Toxicol 2004), hence affecting immune cell trafficking. **Genetical** approaches for breast cancer induction include the targeting of **growth factors** (e.g., TGF- α , Erbb2, IGF2) or their inhibitors (e.g., p53, TGF- β , BRCA1/2) which are well-known to control the migration of immune cells (IGF2: Hartnell et al., J Immunol 2004; p53: Sutton et al., J Am Soc Nephrol 2013; TGF- β ; Phillips et al., J Submicrosc Cytol Pathol 1993, Walshe et al., ATVB 2009; BRCA1: Mak et al., Nat Immunol 2000; BRCA2: Reislander et al., Nat Commun 2019), change the phenotype of immune cells (TGF- α : Walz et al., Leukemia 1995; TGF- β : Fridlender et al., Cancer Cell 2009), and regulate the synthesis of cytokines and chemokines (Erbb2: Madson et al., Am J Pathol 2006; IGF2: Yang et al., J Allergy Clin Immunol 2014), ultimately affecting immune cell trafficking. In addition, targeting **intracellular signaling pathways** for tumor induction (e.g., WNT, c-Myc, FGF, Notch-4, ras) alters the production of pro-inflammatory factors (WNT:

Lengfeld et al., PNAS 2017; c-Myc: Konsavage et al., Dig Dis Sci 2013; FGF: Qi et al., Mol Med Rep 2019; Notch-4: Xia et al., J Allergy Clin Immunol 2018; ras: Zahng et al., Eur J Pharmacol 2014), thus affecting immune cell trafficking. Moreover, using viral oncogenes (e.g., SV40) for tumor induction also modulate the production of cytokines such as TNF (Tokorodani et al., Biol Pharm Bull 2020), hence affecting immune cell trafficking.

Consequently, we employed the 4T1 breast cancer model in immunocompetent mice which is well-established to study endogenous immune cell responses in mammary malignancies.

Comment 4.) Some typos such as microvacular (page6) and their (which should be "the" on page 7)...

Answer: Thank you very much. Corrected.

Referee #3

General comment: The findings are interesting but it is very technical without clear explanation for general readership. It need expanding of the text especially the results for this to becomes more clear to a reader who does not have immunological expertise or imaging of the particular technology used by the authors.

In this manuscript, Uhl et al. present evidence for the role of urokinase-type plasminogen activator (uPA)-plasminogen activator inhibitor-1 (PAI-1) heteromers in breast cancer. The authors demonstrate the potential for clinical relevance by associating uPA and PAI-1 protein levels with both extravascular and intravascular neutrophils, from breast cancer cohort patients, that were diagnosed with G1 breast cancer. uPA-PAI-1 heteromers are part of the fibrinolytic system, impacting cell adhesion and migration, angiogenesis, signal transduction and apoptosis. uPA-PAI-1 can recruit both macrophages and neutrophils and tumor derived uPA-PAI-1 protein expression, is known to impact breast cancer and influence metastasis. The authors utilized a mouse peritoneal assay(s), along with multi-channel flow cytometry, to track neutrophil motility in response to uPA-PAI-1 heteromers treatment. In response to uPA-PAI-1 exposure neutrophils, increased transmigration activity, in the intertumoral space thus promoting early tumor growth and metastasis. In vivo, uPA-PAI-1 exposure increases the number of both macrophage and neutrophils into the peritoneal cavity, neither B and or T lymphocytes were similarly recruited after treatment. The activity of uPA-PAI-1 does not depend on activating endothelial cells expression of either ICAM and or VCAM, for transmigration activity. Rather uPA-PAI-1 heteromers activity increase ICAM and VCAM expression in a neutrophil specific manner through the supposed activity of TNF. The authors present data that increase TNF production arises from the macrophages in response to uPA-PAI-1 exposure. This would suggest that uPA-PAI activation is upstream of TNF as it increases ICAM and VCAM expression in neutrophils but not in the endothelial cells. Moreover, uPA-PAI-1 activity in vivo, that influenced the NE neutrophils subtype rather than MMP9 and or VEGF subtype. The NE neutrophils stimulate 4T1 proliferation in vitro and in vivo uPA-PAI-1 inhibitor (WX340) decreased neutrophil recruitment delaying 4T1 cells tumor growth and decreasing metastasis.

In general, the manuscript has very intriguing results and mostly well control. Description(s) of the experimental methodology was often lacking in details within the text causing one to pause and ask was the specific assay been conducted in vivo or in vitro? Description of muscle cremaster models using the Ly-6G+ CX3CR-1- cells is poorly described in this manuscript it is referenced as published, should be introduce for general readership. Some results where conflated in one experimental system to conclude similar findings in a second experimental system and suggesting the impact on cellular activity was the same without providing raw data for independent assessment of these conclusions. Tumors biology lack

tracking of tumor growth curves either daily or weekly. Histomorphogenic analysis of tumors and metastasis lesion was absent to ensure the arguments that this was due to less neutrophil recruitment. Validation of the IHC analysis to make conclusion in regards blocking growth and or preventing metastasis was not convincing. In all the manuscript has made a very important observation(s) and conclusions that merit considerations for publication. This would require major editing and response to several major and minor points so that impact of the work is broadly accepted.

Answer: The authors appreciate the very positive evaluation of their manuscript.

Comment 1.) Fig. 1B it is not clear why the author depleted the neutrophils with Ly-6B yet identify the neutrophils as Cd11bhi/GR-1hi/CD115 neg, the raw FACS data needs to be shown. Ly-6B is a general granulocyte marker also found in eosinophils and basophils. Ly-6B is also found in activate macrophages and uPA-PAI-1 activated macrophage recruitment to the peritoneal cavity. So it is misleading to called it a "neutrophil depletion" when it also depletes macrophages as shown by the figure 1A.

Answer: Thank you very much for this comment. For the depletion of circulating neutrophils, we made use of a well-established and frequently employed approach with monoclonal anti-Ly-6G antibodies. According to previously published protocols, mice received intra-venous injections of anti-Ly-6G monoclonal antibodies (clone 1A8; 150 µg; 24 h and 6 h prior to induction of inflammation; BD Biosciences, San Jose, CA, USA) which specifically depletes neutrophils (Daley et al., J Leukoc Biol 2008) by Fc-dependent opsonization and phagocytosis of the antibody-bound cells (Bruhn et al., Results Immunol 2016). This point is now clearer in the revised version of our manuscript (p. 26, 1st paragraph).

2.) The neutralizing assay is not detailed in M&M and which antibody was used to do the depletion.

Answer: The experimental approach to deplete neutrophils is now described in more detail in the Materials and Methods section of the revised version of our manuscript (p. 26, 1st paragraph).

3.) Fig 1C As this model is heavily dependent on many of its I conclusion on microscopy imaging it is quite unclear why the cells are label Green (GR1 positive) when utilizing the Ly-6G+ GFP- using the Cx3CR-1GFP+ mice. This highlight the difficult the author creates by not describing their model for this particular manuscript with little details and relying on previous

published work to call the cells neutrophils. This manuscript should stand alone without having to go read another manuscript.

Answer: According to the reviewer's suggestion, technical details regarding our multi-channel *in vivo* microscopy experiments in CX₃CR-1^{+GFP} ('monocyte-reporter') mice are now described in more detail in the revised version of our manuscript. Briefly, CX₃CR-1^{+GFP} mice exhibit GFP^{low} classical monocytes and GFP^{high} non-classical monocytes. In the cremaster muscle of male CX₃CR-1^{+GFP+} mice, neutrophils were identified as Ly-6G⁺ CX₃CR-1⁻ cells, classical monocytes as Ly-6G⁻ CX₃CR-1^{low} cells, and non-classical monocytes as Ly-6G⁻ CX₃CR-1^{high} cells upon intravenous injection of PE-labeled, non-depleting monoclonal anti-Ly-6G antibodies as reported elsewhere (e.g., Zuchtriegel et al., *Arterioscl Thromb Vasc Biol* 2015). This point is now clearer in the revised version of our manuscript (p. 6, 1st paragraph; p.18, 2nd paragraph; p. 23, 3rd paragraph).

4.) A single image is not representative of the quantified data shown for adherence and transmigration assays for either the macrophages and or the neutrophils the data is not impressive showing two cells.

Answer: According to the reviewer's suggestion, we now also show representative *in vivo* microscopy images from the unstimulated control group in the revised version of our manuscript (**Fig. 1c**).

5.) Fig 2a. Cremasteric assay no reference provide in text for this manuscript. Immunofluorescent green for UPA and PAI are they labels with green tags, is unclear. Upon TNF treatment recruitment of neutrophils are pink and green the protein moved it is just not clear at all or are the uPA-PAI-1 heterodimerization occurring in the neutrophils or another cell subtype, this is not at all clear.

Answer: In these experiments, male WT mice received an intrascrotal injection of TNF. 6 hours later, both cremaster muscles were harvested and uPA or PAI-1 protein (shown in green) were visualized separately *ex vivo* by immunostaining and confocal laser scanning microscopy. In addition, postcapillary venules (CD31⁺ cells; shown in red) and Ly-6G⁺ neutrophils (shown in purple) are depicted in these images, respectively (scale bar: 50 μm). Here, uPA and PAI-1 were detected in the perivenular space of inflamed tissue as well as – to a lesser extent – on the microvascular endothelium of postcapillary venules. However, we cannot clearly state to what extent intracellular and extracellular uPA-PAI-1 heteromerization occurs in inflamed tissue. This is now clearer in the revised version of our manuscript and a reference to these findings is provided in the manuscript text (p. 13, 2nd

paragraph; Figure legend 2).

6.) *Fig 2C an in vitro assay with RAW 264.7 exposed to uPA-PAI-1 heteromers is not the same thing as macrophages derived from an in vivo source. Especially if the arguments are that macrophages are been recruit to assist the neutrophils too attach and transmigrate. As they provide the chemokines and our TNF production as the source to elicit such action in vivo, that is transmigration.*

Answer: According to the reviewer's suggestion, we now also show that recombinant murine uPA-PAI-1 heteromers induce the release of TNF from peritoneal macrophages isolated from WT mice (**Appendix Fig. S2a**; p. 7, 1st paragraph).

7. *Fig2D results using an endothelial in vitro assay vs what was shown in Fig2B in vivo are not consistent, in one assay the uPA-PAI-1 heteromers increase ICAM/VCAM but in the other I assay it does not? If the author were attempting to imply that the in vivo assay requires macrophage recruitment for uPA-PA-1 to the endothelial cell for both ICAM/VAM to be up regulated only in vivo but not in vitro this was not clear at all in the text or the explanations of the results. Therefore, culture of the bEND.3 with the RAW cells would potentially address this issue when exposed to either TNF and or uPA-PAI-1 heteromers.*

Answer: We appreciate the suggestion of this reviewer to provide further evidence that uPA-PAI-1-activated macrophages are able to induces the surface expression of ICAM-1/CD54 and VCAM-1/CD106 in endothelial cells. To this end, we performed a novel set of experiments in which we measured the expression of ICAM-1/CD54 and VCAM-1/CD106 in bEND.3 endothelial cells co-cultured with RAW macrophages exposed to saline, TNF, or uPA-PAI-1. Our findings clearly support our previous data, indicating that uPA-PAI-1-primed macrophages activate microvascular endothelial cells (**Appendix Fig. S2b**; p. 7, 1st paragraph).

8. *Fig1E require images for independent assessment that cell is specifically interacting with one cell surface receptor but not the other.*

Answer: In addition to the quantitative flow cytometry data, confocal microscopy images of primary mouse neutrophils exposed to recombinant murine uPA-PAI-1 heteromers or PBS binding to ICAM-1/CD54-Fc or VCAM-1/CD106-Fc are now provided in the revised version of our manuscript (**Appendix Fig. S2c**). In these novel images, enhanced and clustered binding of neutrophils to ICAM-1/CD54, but not to VCAM-1/CD106, is visualized (p. 7, 2nd paragraph).

9. *Fig EV4 needs to show IHC results of the neutrophil population in G1, G2 and G3 breast cancer samples. For independent assessment of authors conclusions.*

Answer: According to the reviewer's suggestion, representative images of neutrophil infiltration in G1, G2, and G3 breast cancer samples is now shown in the revised version of our manuscript (**Fig. EV4**).

10. *Figure 3A a single IHC for uPA-PAI-1 is not valid for this analysis and will require showing of multiple samples with wider range of the tumor showing.*

Answer: According to the reviewer's suggestion, more representative images of neutrophil infiltration in breast cancer samples is now shown in the revised version of our manuscript (**Fig. EV4**; see above).

11. *The author do not address the expression for uPA-PAI-1 has been shown that can also come from human tumors and not just the infiltrating neutrophils.*

Answer: We appreciate the notion of this reviewer that not only neutrophils release uPA and PAI-1, but also breast cancer cells. This point is now clearer in the revised version of our manuscript (p. 11, 2nd paragraph; **Appendix Fig. S3a**).

12. *Figure 3b the cohort analyzed for the METABRIC data set need to be clarify as to which subtype of breast cancer were analyzed. Where there any differences between Luminal A and or B, Her 2 and or basal/TNBC subtypes. No p-value and or HR are shown diminishing the significance of the study. Simple correlation are not indicate of clinical relevance.*

Answer: We thank the reviewer for this essentially important comment and apologize that this was not mentioned in the original version of our manuscript. P-values and HR are now shown in **Fig. 3b** and **Fig. EV5b**. Further, cross-tabulation and Chi-squared analysis revealed no significant association between high RNA expression of PLAU or SERPINE1 in the tumor and the molecular breast cancer subtype as defined by the 3-gene-classifier. Accordingly, we conclude that the significantly impaired overall survival of stage 0-1 breast cancer patients with PLAU or SERPINE1 high expressing tumors is not confounded by enrichment of a distinct molecular subtype. These results have been added to the revised version of our manuscript in form of a mosaic plot in **Fig. EV5c**.

13. *Figure 3d it is not clear why the inhibitor WX-340 was not added to the neutrophils plus 4t1 cells or when NE inhibitor was added to show combinatorial effects.*

Answer: We appreciate the reviewer's suggestion to analyze the effect of compound WX-340 on the proliferation of 4T1 breast cancer cells in more detail. In a novel set of experiments, application of WX-340 did not significantly alter the proliferation 4T1 tumor cells exposed to the supernatant of uPA-PAI-1-primed neutrophils. Similarly, additional application of WX-340 did not significantly change the proliferation of 4T1 tumor cells upon inhibition of NE. These data are now included in the revised version of our manuscript (**Appendix Fig. S2d**).

14. Fig4a not clear what cells are been used either the text or the figure legends. Similar for Fig4b not clear if this is in the context of the tumor in vivo model or not.

Answer: In **Fig. 4a**, the effect of WX-340 on the binding of recombinant mouse uPA protein to PAI-1 protein was quantitatively assessed *in vitro* using ELISA analyses. In **Fig. 4b**, the effect of WX-340 on intravascular rolling and firm adherence as well as on transmigration of neutrophils and classical monocytes to the inflamed perivascular tissue was analyzed by multi-channel *in vivo* microscopy in the mouse cremaster muscle upon intra-scrotal injection of TNF. This point is now clearer in the revised version of our manuscript (Figure legend 4; p. 11, 1st paragraph).

15. Fig4C the staining for uPA and or PAI-1 are unimpressive and do not appear to be specific but rather a wide expression pattern is shown to most every cell.

Answer: According to the reviewer's suggestion, we now provide images in higher magnification in the revised version of our manuscript (**Appendix figure S3a**) which support previous findings on the wide expression of uPA and PAI-1 in both breast cancer and immune cells.

16. The author does not rule out the effects on lack of macrophage recruitment to the tumor biology his needs to be addressed.

Answer: Thank you very much for this suggestion. Circulating classical monocytes are recruited to the tumor environment, where these immune cells differentiate into macrophages (Zhou et al., *Front Oncol* 2020; Patel et al., *J Exp Med* 2017; Tak et al., *Blood* 2017; Evren et al., *Immunity* 2021). In the present study, we demonstrate that uPA-PAI-1 heteromers potently recruit neutrophils and, in turn, classical monocytes from the microvasculature to the perivascular tissue. To evaluate the relative contributions of the recruitment of these two immune cell subsets for tumor growth, we performed a completely novel set of experiments. Here, we show that antibody-mediated depletion of neutrophils severely compromised tumor growth in 4T1 breast cancer, whereas antibody-mediated depletion of classical monocytes

exhibited a lower, but significant tumor-suppressive effect than depleting neutrophils. These novel data are now included in the revised version of our manuscript (**Appendix Fig. S3c; Appendix Tab. S2**).

17. Fig4D tumor growth kinetics are lacking along with histomorphogenic analysis of tumors.

Answer: According to the reviewer's suggestion, tumor growth kinetics are now shown in the revised version of our manuscript (**Appendix Fig. S3b**).

18. Figure 4F effects on metastasis by 4T1 cells is very robust to multiple organs yet these results cannot be independently asses by the data shown.

Answer: In our experiments, we employed 4T1 tumor cells which were transfected with a tomato fluorescent protein. This approach enabled us to quantitatively analyze the number of metastasized tumor cells 14 days after tail vein injection of these cells in whole organs by multi-channel flow cytometry which is supposed to be more accurate than the histomorphogenic analysis of tissue sections.

To enable the more independent assessment of the effect of WX-340 on tumor metastasis in lungs and brain, we provide representative images of immunohistochemically stained tissue sections in the revised version of our manuscript (**Appendix Fig. S4b**).

19. IHC and HE comparison are requires and timing of the experiment metastasis is also lacking.

Answer: As mentioned above, we now also provide representative images of immunohistochemically stained tissue sections as well as a schematic illustration of the experimental protocol of our tumor metastasis experiments (**Appendix Fig. S4a, b**).

12th Mar 2021

Dear Prof. Reichel,

Thank you for the submission of your revised manuscript to EMBO Molecular Medicine. We have now received the enclosed reports from the two referees who re-reviewed your manuscript. As you will see, they are now supportive of publication, and I am therefore pleased to inform you that we will be able to accept your manuscript, once the following minor points will be addressed:

1) Referees' comments:

Please address the remaining point from referee #1 regarding the reference to add and discuss.

2) Main manuscript text:

- Please answer/correct the changes suggested by our data editors in the main manuscript file (in track changes mode). This file will be sent to you in the next couple of days. Please use this file for any further modification.
 - Conflict of interest: please provide the full sentence "The authors have no conflict of interest to declare".
 - Material and methods:
 - o Cells: please indicate whether the cells were tested for mycoplasma contamination.
 - o Antibodies: please indicate the dilutions used in your study.
 - o Human samples: please include a statement confirming that written informed consent was obtained from all subjects and that the experiments conformed to the principles set out in the WMA Declaration of Helsinki and the Department of Health and Human Services Belmont report. Please identify the committee approving the protocol.
 - Statistics: Please indicate in the figures or in the legends the exact $n=$ and exact $p=$ values along with the statistical test used. You may provide these values as a supplemental table in an Appendix file.
 - After the Material and Methods section, please replace "Data sharing" by a "Data availability section": primary datasets produced in this study need to be deposited in an appropriate public database (see <https://www.embopress.org/page/journal/17574684/authorguide#dataavailability>). If no new dataset was generated, please indicate: "This study includes no data deposited in external repositories".
- Please remove "For original data, please contact christoph.reichel@med.uni-muenchen.de"

3) Figures and tables:

- Please upload the figures separately as independent files (without legends). Please make sure the figures are big enough to be fully visible/readable.
- The manuscript text refers to figure EV9, please clarify.

4) Checklist: please provide information in sections E/11-12. Please also provide information in section F/18. Regarding section F/19, please clarify to which dataset you are referring to.

5) Thank you for providing 'The paper explained'. I slightly edited the text to shorten it, please let me know if you agree with the following:

PROBLEM: Breast cancer is the most common oncological disorder in women worldwide. High intratumoral levels of heteromers of the serine protease urokinase-type plasminogen activator (uPA)

and its inhibitor plasminogen activator inhibitor-1 (PAI-1) predict impaired survival and treatment response already in early stages of breast cancer. Although these single proteins are well known to control tissue perfusion by regulating clot formation as key components of the fibrinolytic system, the pathogenetic role of this protein complex in breast cancer remains obscure.

RESULTS: Utilizing patient data and different syngeneic mouse models of breast cancer, we demonstrate that heteromerization of uPA and PAI-1 multiplies the potential of the single proteins to attract pro-tumorigenic neutrophils. To this end, tumor-released uPA-PAI-1 utilizes the very low density lipoprotein receptor and intracellular mitogen-activated protein kinases to initiate a pro-inflammatory program in perivascular macrophages in the proximity of malignant tumors. This enforces neutrophil trafficking to cancerous lesions and skews these immune cells towards a pro-tumorigenic phenotype, thus supporting tumor growth and metastasis. Blockade of uPA-PAI-1 heteromerization by a novel small-molecule inhibitor interfered with these events and effectively prevented tumor progression.

IMPACT: Our findings identify a therapeutically targetable, hitherto unknown interplay between hemostasis and innate immunity that drives breast cancer progression. As a personalized immunotherapeutic strategy, blockade of uPA-PAI-1 heteromerization might be particularly beneficial for patients with highly aggressive uPA-PAI-1high tumors.

6) Thank you for providing a 'For more information' section. Please note that the English webpage was not accessible, however the German one was (<https://www.femtelle.de/>).

7) Thank you for providing a nice synopsis picture. Please upload it separately as a png or jpeg file 550px wide x 400px high.

8) As part of the EMBO Publications transparent editorial process initiative (see our Editorial at <http://embomolmed.embopress.org/content/2/9/329>), EMBO Molecular Medicine will publish online a Review Process File (RPF) to accompany accepted manuscripts.

This file will be published in conjunction with your paper and will include the anonymous referee reports, your point-by-point response and all pertinent correspondence relating to the manuscript. Let us know whether you agree with the publication of the RPF and as here. Please note that the Authors checklist will be published at the end of the RPF.

I look forward to receiving your revised manuscript.

Yours sincerely,

Lise Roth

Lise Roth, PhD
Editor
EMBO Molecular Medicine

***** Reviewer's comments *****

Referee #2 (Comments on Novelty/Model System for Author):

The technical quality of the present manuscript is high and the employed model systems are adequate. The novelty of the described findings are medium as the effects of WX-340 on breast cancer progression have already been described by others (Anti-tumor and anti-metastatic activity of WX-340 a highly specific uPA-inhibitor in the rat BN-472 mammary carcinoma model. Buddy Setyono-Han, Anneliese Schneider, Mieke Timmermans, Anieta Sieuwerts, Francesco Blasi, Wolfgang Schmalix and John Foekens Cancer Res May 1 2007 (67) (9 Supplement) 5615;) An application of WX-340 in therapy needs lengthy approval thus a direct medical impact is not given.

Referee #2 (Remarks for Author):

The authors have addressed most of the concerns about the previous version of the manuscript. Additional in vivo validations would be reconfirming but might exceed the scope of the current work and can be followed up in the future. The previous work on WX-340 in breast cancer (Anti-tumor and anti-metastatic activity of WX-340 a highly specific uPA-inhibitor in the rat BN-472 mammary carcinoma model.

Buddy Setyono-Han, Anneliese Schneider, Mieke Timmermans, Anieta Sieuwerts, Francesco Blasi, Wolfgang Schmalix and John Foekens Cancer Res May 1 2007 (67) (9 Supplement) 5615) needs to be cited and should be discussed.

Referee #3 (Comments on Novelty/Model System for Author):

The authors have addressed my previous concerns and have improved the quality of manuscript substantially, improved figure and quality of data has improved. For general readership the text issues from previous review have been addressed.

Referee #3 (Remarks for Author):

The authors have address all concerns from this reviewer. Th authors greatly improve both the quality of results and readership.

The authors performed the requested editorial changes.

25th Mar 2021

Dear Prof. Reichel,

Thank you for sending the revised files. I looked at everything and all is fine. I am thus very pleased to accept your manuscript for publication in EMBO Molecular Medicine!

Your manuscript will now be sent to our publisher to be included in the next available issue.

Please read below for additional important information regarding your article, its publication and the production process.

Congratulations on a nice study!

Yours sincerely,

Lise Roth

Lise Roth, Ph.D
Editor
EMBO Molecular Medicine

Corresponding Author Name: Christoph Reichel

Journal Submitted to: EMBO Mol Med

Manuscript Number: EMM-2020-13110